# Mapping Functional Urban Green Types Using High Resolution Remote Sensing Data

**Jeroen Degerickx** [1,2,*], **Martin Hermy** [1] **and Ben Somers** [1]

1   Division of Forest, Nature and Landscape, KU Leuven, Celestijnenlaan 200E, 3001 Leuven, Belgium;
    martin.hermy@kuleuven.be (M.H.); ben.somers@kuleuven.be (B.S.)
2   VITO Remote Sensing, Boeretang 200, 2400 Mol, Belgium
*   Correspondence: jeroen.degerickx@vito.be

**Abstract:** Urban green spaces are known to provide ample benefits to human society and hence play a vital role in safeguarding the quality of life in our cities. In order to optimize the design and management of green spaces with regard to the provisioning of these ecosystem services, there is a clear need for uniform and spatially explicit datasets on the existing urban green infrastructure. Current mapping approaches, however, largely focus on large land use units (e.g., park, garden), or broad land cover classes (e.g., tree, grass), not providing sufficient thematic detail to model urban ecosystem service supply. We therefore proposed a functional urban green typology and explored the potential of both passive (2 m-hyperspectral and 0.5 m-multispectral optical imagery) and active (airborne LiDAR) remote sensing technology for mapping the proposed types using object-based image analysis and machine learning. Airborne LiDAR data was found to be the most valuable dataset overall, while fusion with hyperspectral data was essential for mapping the most detailed classes. High spectral similarities, along with adjacency and shadow effects still caused severe confusion, resulting in class-wise accuracies <50% for some detailed functional types. Further research should focus on the use of multi-temporal image analysis to fully unlock the potential of remote sensing data for detailed urban green mapping.

**Keywords:** multispectral; hyperspectral; LiDAR; vegetation monitoring; ecosystem services; object-based image analysis; data fusion; remote sensing; machine learning; Random Forest models

## 1. Introduction

Worldwide, urban areas are faced with major challenges imposed by rapid urbanization trends and the increased occurrence of extreme weather events due to climate change [1]. In order to safeguard the quality of urban life, cities need to be designed and managed in a smart and (more) sustainable way. In this respect, urban green represents an important tool due to the many ecosystem services (i.e., direct or indirect benefits to human society [2]) it may provide, including provisioning (e.g., food production), regulating (e.g., mitigating urban heat waves, floods and air pollution), cultural (e.g., recreation) and supporting (e.g., biodiversity, pollination) services [3,4]. Quantifying ecosystem services provided by urban green in a spatially explicit way, or the production of ecosystem service maps, has been proposed as a valuable tool in support of sustainable urban planning, development and policy making [4–7]. Indeed, such maps could be used to identify problematic urban zones featuring a lack of a particular or multiple ecosystem services which should subsequently be prioritized in urban development plans [8–10]. Moreover, by generating ecosystem service maps for different urban planning scenarios, informed and sustainably sound decisions can be made to ensure high environmental quality in our future cities [11,12]. Lastly, due to the ever growing knowledge on the link between particular plant traits and ecosystem services, such maps can assist urban green managers to design green spaces

which are not only aesthetically appealing, but which also maximize the diversity and magnitude of ecosystem services provided [13].

A frequently used way of mapping ecosystem services, referred to as the value-transfer approach, relies on the combination of a land cover map and a pre-defined scoring table in which each of the land cover classes is assigned an ecosystem service score, which in itself can consist of a simple ranking or a more advanced quantitative score [8,14–17]. Most of these ecosystem service mapping efforts however focus on broad land cover classes (in ecosystem services literature referred to as service providing units, e.g. forest, wetland, garden, park, allotment, agricultural land [5]), failing to capture the important effects of the specific type, properties and context of urban green on ecosystem service provisioning [5,15,18,19]. On the other hand, several more detailed typologies of urban green have been suggested, each designed for a specific application, e.g., biodiversity monitoring [20,21], land use [22,23], urban climate [24], urban hydrology and cooling [15] and management of public urban green (e.g. urban green administration of the city of Brussels, oral communication). In order to get an integrated yet detailed view on ecosystem services provided by urban green, we suggest the construction of a functional urban green typology, i.e., a typology solely based on the main functions and services of urban green and taking into account vegetation type, relevant properties and contextual information.

Aside from a functional urban green typology, an operational mapping workflow is required to effectively monitor these detailed urban green types at a city-wide scale. In this paper, rather than relying on labor- and time-consuming field inventories, we explore the potential of remote sensing data acquired from airplanes and satellites, in combination with state-of-the-art image processing techniques, for mapping of functional urban green types. In particular, our main focus is on the use of optical remote sensing data, measuring the reflectance of solar light on the earth's surface within the visible (VIS; 0.4–0.7 μm), near-infrared (NIR; 0.7–1.25 μm) and short-wave infrared (SWIR; 1.25–2.5 μm) domains of the electromagnetic spectrum (Figure 1). As each object interacts differently with different parts of this spectrum, the reflected signal can be used as a basis for (urban) land cover mapping (Figure 1, [25]). Due to the very subtle differences in spectral reflectance between different vegetation types and between individual species, detailed vegetation mapping generally requires the use of hyperspectral sensors (in which reflectance is measured in high detail using many, narrow and contiguous spectral bands; Figure 1 [26–31]). Although the technological advances and number of applications for hyperspectral sensors on board of satellites [32] and UAVs (unmanned aerial vehicles, or drones [33]) are slowly increasing, most hyperspectral imagery today is captured using airplanes, generating detailed imagery with a spatial resolution (pixel size) of 2–15 m. Due to the high spatial complexity and heterogeneity of urban areas however, airborne hyperspectral data is typically characterized by a high share of mixed image pixels, i.e., pixels containing more than one land cover class, in turn severely complicating further analysis [34,35]. Despite a growing number of subpixel mapping approaches, e.g., [36,37], detailed urban green mapping remains challenging due to the high spectral similarity amongst individual urban green types [27].

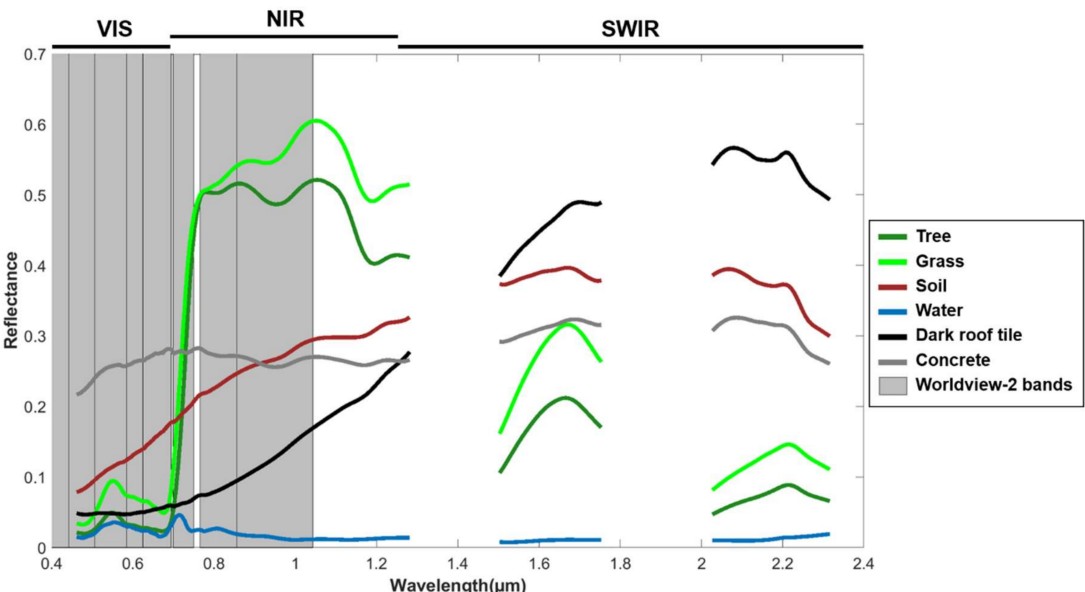

**Figure 1.** The difference between hyperspectral signals (plotted as individual lines) and multispectral signals. For the latter, only the spectral band limits have been plotted as grey rectangles. The multispectral sensor (in this case Worldview-2) only records a single reflectance value per band. From the examples plotted here, it becomes clear that spectral data can be used to identify objects. The different parts of the spectrum have been indicated on top of the graph, where VIS = visible (0.4–0.7 μm), NIR = near-infrared (0.7–1.25 μm) and SWIR = short-wave infrared (1.25–2.5 μm).

Over time, many approaches have been suggested to allow for more detailed urban mapping. Firstly, fusion of spectral data with LiDAR (Light Detection and Ranging) data has been successfully applied in urban areas for land cover mapping [38–42], tree species classification [43], urban green mapping [44], detection of invasive shrub species [45] and tree health estimation [46] due to the high complementarity between spectral data and spatially very detailed structural information derived from 3D LiDAR data. Secondly, hierarchical (or stratified) classification approaches (i.e., classification done at multiple thematic levels, where each level is used as a constraint to map the next, more detailed level) have been shown to increase the mapping accuracy of detailed land cover classes [47–49]. Thirdly, Object-Based Image Analysis (OBIA), in which similar pixels are grouped into homogeneous image objects prior to classification, represents another promising technique [50]. By using objects rather than pixels, additional information (i.e., size, shape and internal variability of image objects) becomes available to the classification algorithm. Although commonly being applied on (airborne) hyperspectral data [51–54], the added value of OBIA becomes most apparent when applied on high spatial resolution multispectral data, allowing distinction between detailed land cover classes based on limited spectral information [55–59]. Here, we will further explore these three analysis techniques, specifically for detailed urban green mapping.

In summary, the overall goal of this study is to develop a framework (typology) and associated workflow based on remote sensing data for accurate mapping of functional urban green types. By assessing the potential of various remote sensing data sources, i.e., airborne hyperspectral data, high-resolution multispectral satellite data and airborne LiDAR data (and different combinations thereof), we additionally aim for increased insight into the most relevant input data to be used for detailed urban green mapping.

## 2. Materials and Methods

### 2.1. Functional Urban Green Typology

Urban green can be studied on many different scales, ranging from parcel level (park, garden) to the individual plant scale [6]. As parks and gardens may provide entirely different services depending on their composition (e.g., a lawn mainly serves as a playground for kids, whereas a botanical garden is more interesting from an ecological, educational and scientific point of view [18]) and individual plants may serve other purposes depending on their context (e.g., a row of street trees as part of an ecological network, versus a solitary tree for ornamental purposes), we decided to focus on urban green elements as the main unit of our typology. An urban green element is defined here as an assemblage of individual plants together providing similar functions and services.

Based on a literature review, combined with in-house expert knowledge, we identified key plant properties affecting ecosystem service provisioning, including all four ecosystem service categories (provisioning, regulating, cultural and supporting services [2]). Using these insights, we categorized urban green elements into a total of 23 functional urban green types and provided a qualitative score on the contribution of each type to the most relevant urban ecosystem services (Table 1). Functional urban green types were categorized into three main categories, i.e., tree, shrub and herbaceous plants. Due to their large size and leaf area compared to other urban green elements, trees are known to excel at providing regulating ecosystem services [60–62]. Further distinction into multiple tree functional types was therefore mainly based on their production potential (food and woody biomass), cultural benefits (potential for recreation and aesthetic value), internal biodiversity and their potential to support more biodiversity. The extent (surface area), structural diversity, spatial configuration (shape, area/edge length, connectivity) and management (frequency of harvesting and human disturbance) were identified as the main factors affecting the provisioning of these specific ecosystem services by urban trees [63–66]. Based on these characteristics, eight tree functional types were defined, ranging from (semi-)natural forests to individual isolated trees (Table 1). Precise definitions of these functional urban green types were based on expert knowledge and local good practices and guidelines for urban green management. Specifically in terms of regulating services, leaf phenology (evergreen/deciduous), leaf type (broadleaf/coniferous) and tree size were found to be crucial factors [60–62,67]. Although not explicitly included in our typology, we highly recommend these tree characteristics to be used as supplementary information to further refine any ecosystem service assessment of urban trees.

Due to a general lack of scientific literature specifically focusing on ecosystem service provisioning by shrubs in an urban context, largely the same reasoning as used for trees was applied. Three functional types were defined based on a combination of extent and spatial configuration (Table 1). Due to their size and compact shape, large scrub patches can significantly contribute to regulating ecosystem services and provide valuable habitats for various animal species. Hedges are a specific type commonly used in urban areas as noise and privacy barriers, but at the same time present habitat opportunities for smaller animals, both vertebrates and invertebrates [68]. Finally, individual or small groups of shrubs, mainly planted for ornamental purposes in parks and gardens, were treated as a separate functional type. As for trees, leaf phenology and leaf type constitute important complementary information for detailed assessment of ecosystem services.

**Table 1.** Functional urban green typology proposed in this study. For each type, an indication of its relevance for several provisioning, regulating, cultural and supporting ecosystem services is included (X = important contribution; (X) = low contribution; blank = (almost) no contribution). Functional types that are not being covered in the remote sensing based mapping part of this study have been greyed out.

| Functional Urban Green Type | Definition | Provisioning | | Regulating | | | | | | Cultural | | Supporting | |
|---|---|---|---|---|---|---|---|---|---|---|---|---|---|
| | | Food | Biomass | Air Purification | Micro-Climate | C Sequestration | Water: Quantity | Water: Quality | Noise and Visual | Recreation | Visual Attractiveness | Internal Biodiversity | Supporting Biodiversity |
| **TREES** [*1] [*2] | | | | | | | | | | | | | |
| Forest | Area dominated by densely planted or naturally grown trees. Canopy is closed, except for forests in early succession stage. The ecological function is more important compared to the production function. | | X | X | X | X | X | X | X | X | X | X | X |
| Tree plantation | Trees planted at regular and nearly constant intervals from one another, usually with herbaceous or grassy undergrowth. Canopy is not necessarily closed. Trees are around the same age and size. Main function is production. | X | X | X | X | X | X | X | X | | | | X |
| Wood verge | A dense mixture of different species of trees and shrubs. Shape is linear; used as a fence next to e.g. roads, watercourses, private property. | | | X | X | X | X | X | X | | | X | X |
| Tree patch | A group of trees together forming a closed canopy. | | | X | X | X | X | X | | | | | X |
| Tree row | Trees planted at regular and nearly constant intervals (3–15 m) in one or multiple rows. Trees are around the same age. Maximum width is 30 m. | | | X | X | X | X | X | X | | X | | X |
| Espalier | Trees (or large shrubs) intensively pruned and guided in a way that all branches occur in one vertical plane. May also occur next to a building facade. | (X) | | (X) | X | (X) | X | X | X | | X | | X |
| Connected solitary tree | A single tree positioned close to other trees (distance smaller than 15 m). | | | X | X | X | X | X | | | X | | X |
| Isolated solitary tree | A single tree positioned in a relatively wide, open space (distance to nearest tree larger than 15 m). | | | X | X | X | X | X | | | X | | |

| Functional Urban Green Type | Definition | Provisioning | | Regulating | | | | | | Cultural | | Supporting | |
|---|---|---|---|---|---|---|---|---|---|---|---|---|---|
| | | Food | Biomass | Air Purification | Micro-Climate | C Sequestration | Water: Quantity | Water: Quality | Noise and Visual | Recreation | Visual Attractiveness | Internal Biodiversity | Supporting Biodiversity |
| **SHRUBS \*1** | | | | | | | | | | | | | |
| Scrub patch | Large surface area covered with shrubs (width >15 m). | | | X | X | X | X | X | X | | (X) | X | X |
| Hedge | A row of shrubs or small trees, planted within 1 m from each other and regularly (once or multiple times per year) sheared. Maximum width is 2 m. | | | (X) | (X) | (X) | (X) | (X) | X | | (X) | | X |
| Group of shrubs | A group of shrubs of less than 15 m wide or a solitary individual, mainly planted for ornamental purposes. | (X) | | (X) | (X) | (X) | (X) | (X) | | | X | | |
| **HERBACEOUS PLANTS** | | | | | | | | | | | | | |
| Lawn | Homogeneous patch dominated by grass species and regularly mown. | | X | | (X) | (X) | | | | X | | | |
| Pasture | Diverse patch dominated by grass species which is grazed by animals. | | | | (X) | (X) | | | | X | X | X | X |
| Meadow | Diverse patch dominated by grass species which is infrequently mown. | | X | | (X) | (X) | | | | X | X | | X |
| Flower bed | Patch planted with herbaceous non-grass species, mainly for ornamental purposes, also including plants planted in pots. | | | | | | | | | | X | X | (X) |
| Tall herb vegetation | Dense herbaceous vegetation of more than 1 m high. | | X | | (X) | X | | | | | | X | X |
| Flower field | Patch dominated by herbaceous non-grass species, natural situation. | | (X) | | (X) | (X) | | | | | X | X | X |
| Water plants | Plants fully living in water, either submerged or near the water surface. | | | | | | | (X) | | | X | (X) | X |
| Arable land | Large land surface used for crop production. | X | | | | (X) | | | | (X) | | | X |
| Vegetable garden | Small-scale farming. Typically, different crops are combined on a small piece of land. | X | | | (X) | | | | | | | (X) | (X) |

| Functional Urban Green Type | Definition | Provisioning | | Regulating | | | | | | Cultural | | Supporting | |
|---|---|---|---|---|---|---|---|---|---|---|---|---|---|
| | | Food | Biomass | Air Purification | Micro-Climate | C Sequestration | Water: Quantity | Water: Quality | Noise and Visual | Recreation | Visual Attractiveness | Internal Biodiversity | Supporting Biodiversity |
| Climbers and plant walls | Climbing or non-climbing plants (partially) covering a wall, with or without additional infrastructure to support the plants. This type also includes plants that spontaneously grow directly on (old) walls. | | | X | X | | X | X | X | | X | | (X) |
| Extensive green roof | Green roof with limited substrate depth (max. 20 cm) dominated by Sedum (leaf succulent) species and possibly other spontaneous herbaceous species. | | | (X) | X | | X | X | | | X | (X) | X |
| Intensive green roof | Green roof with substrate depth >20 cm, containing a mixture of grass, herbaceous plants, shrubs and/or trees. | (X) | | (X) | X | (X) | X | X | | X | X | (X) | X |

*1 Each of the urban green functional types within this category should be further divided according to phenology (evergreen/deciduous) and leaf type (broadleaf/coniferous). *2 Each of the urban green functional types within this category should be further divided according to size (height).

From a functional perspective, herbaceous plants are significantly different from trees and shrubs. Due to their relatively small size, their contribution to regulating services is rather limited [69]. Herbaceous and woody vegetation closely associated with buildings (allowed to grow closely against building façades or on roofs) however represents a notable exception to this general rule of thumb. Many previous studies have shown the regulating benefits of these urban green types, which mainly relate to stormwater management, water purification, improved insulation of buildings and mitigating air pollution [70–76]. Therefore, both façade vegetation and green roofs have been identified as separate functional urban green types (Table 1). Additional distinction among herbaceous urban green types was based on their food and biomass production potential (in turn determined by human management and plant characteristics such as presence of edible plant parts, plant height and growth rate) and their internal plant composition (flowering versus grass plants) and diversity. The latter two characteristics both affect to a large extent their visual appeal [77] and potential to support biodiversity and pollination [78]. A total of twelve functional types dominated by herbaceous plants were defined (Table 1). Food crops, which are gaining more attention in urban areas [79,80], were divided into large-scale agricultural fields and small-scale allotment gardens. The latter are characterized by higher structural and plant diversity, in turn contributing to various other ecosystem services [81]. Grass-dominated types (including lawns, pastures and meadows) were subdivided based on their internal biodiversity and human use, whereas further distinction within flowering plants was made based on size (tall versus low herbs) and degree of human interference (semi-natural flower fields and water plants versus intensively managed flower beds).

## 2.2. Mapping Functional Urban Green Types Using Remote Sensing

### 2.2.1. Study Area, Selection of Functional Urban Green Types and General Classification Approach

The Brussels Capital Region is defined as an administrative region consisting of the city of Brussels together with 18 surrounding municipalities. This region is among the most densely built and intensely used areas for residential, commercial and industrial purposes in Europe [82]. Nevertheless, its total area of green space has been estimated at 8714 ha, or 54% of its total area [83], of which roughly 20% is privately owned [84]. Most urban green is located near the edges of the Capital Region (30%–70% urban green cover), whereas the dense city center only contains around 10% of green space [84]. The exact extent of our study area was dictated by the availability of airborne remote sensing data used in this study and is situated in the eastern part of the Capital Region (Figure 2). This particular area comprises a large diversity of urban structure types, including dense residential zones in the west, sparse residential zones in the east and south and industrial/commercial and more rural areas in the north. Some urban green types defined in our functional typology were not considered in the remainder of this study, either because of their intrinsic dimensions making them nearly impossible to detect using remotely sensed data sources, or because of their limited occurrence within our study area (see greyed-out entries in Table 1).

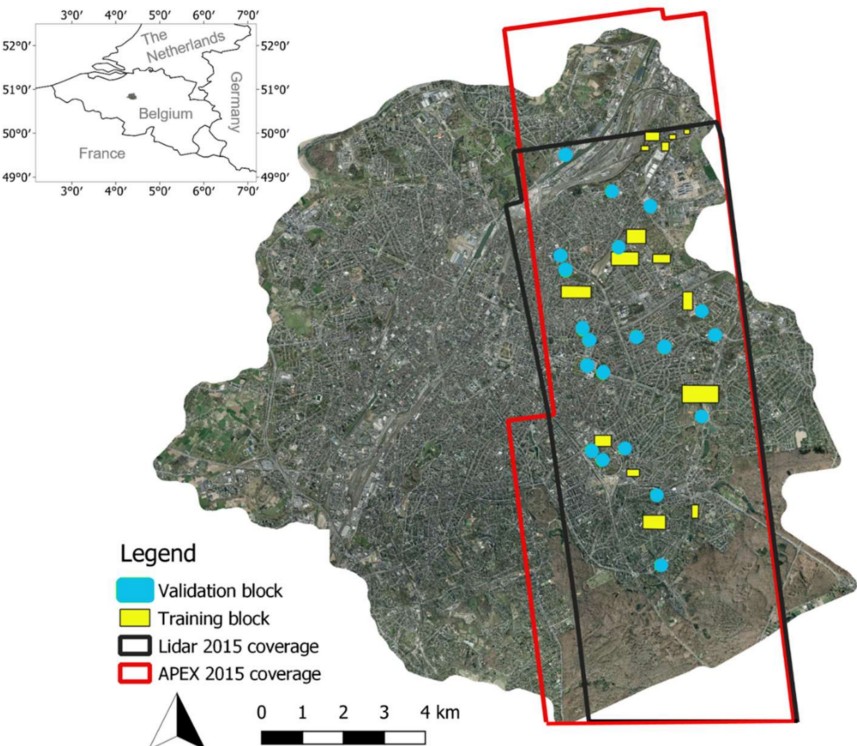

**Figure 2.** Location and extent of remote sensing datasets used in this study relative to the full extent of the Brussels Capital Region, together with the location of training and validation blocks. The size of validation blocks has been exaggerated for visual purposes, its real size amounts to 100 × 100 m. Worldview-2 data was available for the entire Brussels Capital Region.

As can be seen from Table 1, our functional urban green types were defined both in function of plant type (e.g., deciduous tree versus tall herbaceous vegetation) and spatial configuration (e.g., tree row versus solitary tree). Therefore, we opted here for a two-stage classification approach. In a first stage, the different plant types present within the functional urban typology were identified. A hierarchical classification scheme was defined (Table 3). Aside from the usual non-vegetation classes regularly included in urban land cover studies (roofs, pavement, soil and water [37]), cars were explicitly treated separately due to their high abundance and confusion with shrubs. For this first stage, the potential of different datasets and classification approaches was investigated. Classification results were then used to serve as building blocks in a second, rule-based classification approach to make a distinction between patches, rows and individual trees and shrubs. The reader is referred to Section 2.2.4 for a detailed explanation on the classification approach.

2.2.2. Remote Sensing Data

Airborne hyperspectral data was acquired using the APEX sensor on June 30, 2015. The sensor was operated at a flying altitude of 3600 m a.s.l. which resulted in imagery featuring a spatial resolution of 2 m. The APEX sensor covers the spectral range of 400–2500 nm. After removal of water absorption bands, 218 spectral bands remained for further analysis. More information on image pre-processing can be found in [85]. Airborne LiDAR data was collected around the same time in Summer 2015 by Aerodata Surveys Nederland BV. The resulting LiDAR point cloud data featured an average resolution of 15 pts/m$^2$. Finally, a Worldview-2 image covering the entire Brussels Capital Region and captured on July 24, 2016 was put at our disposal by Brussels Environmental Agency (BIM). Worldview-2 consists of eight spectral bands covering the spectral range between 400 and 1050 nm. The raw image data was atmospherically corrected using ATCOR and orthorectified using a 25 cm digital terrain model

in ERDAS Imagine software. Finally, the spectral bands were pan-sharpened in ENVI 5.2 software (Harris Geospatial Solutions) resulting in a pixel size of 0.5 m.

### 2.2.3. Training and Validation Data on Urban Composition and Functional Urban Green Types

Twenty 100 × 100 m validation blocks were delineated using a stratified random sampling approach throughout the study area, thereby ensuring different urban structure types (dense and sparse residential, industrial/commercial and urban green zones) to be sufficiently represented (Figure 2). Due to privacy and accessibility issues, privately owned green areas were avoided as much as possible. Within these validation blocks, land cover and functional urban green types (according to the typology defined in Table 1) were manually mapped during a field visit, visually aided by a 7.5 cm resolution RGB orthophoto acquired in winter 2014. After digitization, a random subsample of objects was selected within each block to serve as training data.

In addition to the validation blocks, fifteen additional blocks were delineated throughout the study area, ranging in size from 1.7 to 38.6 ha, to further complete our dataset of training objects. Rather than mapping land cover in a spatially continuous way as was done for the validation blocks, points were digitally drawn in these areas and labeled based on the same RGB orthophoto and Google Street View. Drawing of points was done with special attention to those land cover classes and urban green types which were underrepresented in the dataset composed by the validation blocks. Table A1 summarizes the sample size of the training dataset per land cover class and compares these to the relative abundances of the classes in the validation blocks.

### 2.2.4. Detailed Classification Approach

In this study we explored the potential of combining hyper- or multispectral data with structural information derived from airborne LiDAR data in an object-based classification approach to produce a detailed land cover map with particular focus on functional urban green types. In essence, we first identified the most useful features for detailed urban green mapping by training several Random Forest models with varying sets of input data. Secondly, we applied the best performing model to our twenty validation blocks to assess its potential to generate spatially continuous land cover maps. Finally, some additional, rule-based classification steps were performed to enhance the final product. Our detailed workflow is essentially comprised of seven parts (Figure 3), described in more detail in the sections below.

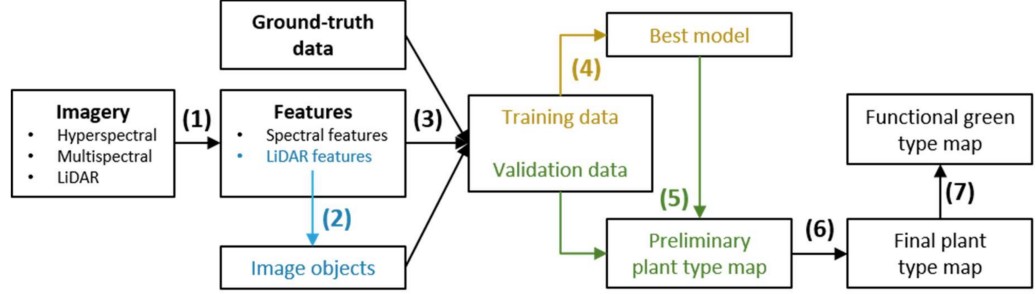

**Figure 3.** Overview of the classification workflow of functional urban green types proposed in this study, including (**1**) spectral and structural feature calculation, (**2**) image segmentation, (**3**) training and validation data generation, (**4**) selection of best model to classify plant type, (**5**) application of best model, (**6**) post-classification correction and (**7**) rule-based classification to discern spatial configuration.

### 2.3. Calculation of Spectral and Structural Features

Hyperspectral datasets typically contain more than 200 spectral bands, often showing high mutual correlations and hence unnecessarily slowing down processing times. Here, we wanted to test whether this information could be summarized without affecting classification accuracy. Two common ways to summarize these data are (1) deriving spectral indices (i.e., ratios of spectral bands known to correlate

with the occurrence or specific property of a particular land cover class) and (2) data transformation specifically aiming at reducing data dimensionality while retaining maximum information content. In this study we calculated a set of eight spectral indices thought to be relevant for urban land cover mapping, i.e., Normalized Difference Vegetation Index (NDVI [86]), Normalized Difference Water Index [87–89], a grass index highlighting the difference between trees and lawn [46], red/green ratio, blue/green ratio and overall brightness (defined as the mean value of all spectral bands). Moreover, we applied a forward Minimum Noise Fraction transformation (MNF [90]) on the APEX bands and retained the first 30 bands based on visual inspection of the resulting eigenvalues. As Worldview-2 data only consists of eight spectral bands, the effect of data reduction was not tested for this dataset. Only NDVI was calculated given its expected relevance for land cover mapping.

The 3D LiDAR point cloud data was converted into a set of 2D features potentially useful for land cover and urban green classification. Aside from height above ground level (normalized digital surface model; nDSM) and intensity, which represent the most frequently used LiDAR features in land cover classification [91], an additional feature related to the permeability of objects was adopted from [92]. This feature, termed treeIndex here, is based on the difference between first and last LiDAR returns and facilitates the differentiation between trees and buildings [46]. All of these features were computed using OPALS software at a resolution of 25 cm, capped off at certain thresholds to remove outliers (i.e., any value above the threshold is set to the threshold value) and scaled between 0 and 1. Two versions of nDSM were created using two different capping thresholds of respectively 15 (nDSM1) and 3 m (nDSM2). Whereas nDSM1 more relates to the actual height of the objects, nDSM2 specifically highlights small height variations, thereby increasing the detection rate of low objects (e.g., hedges, low shrub; Figure 4). Capping thresholds for intensity and treeIndex amounted to 500 and 3 m respectively.

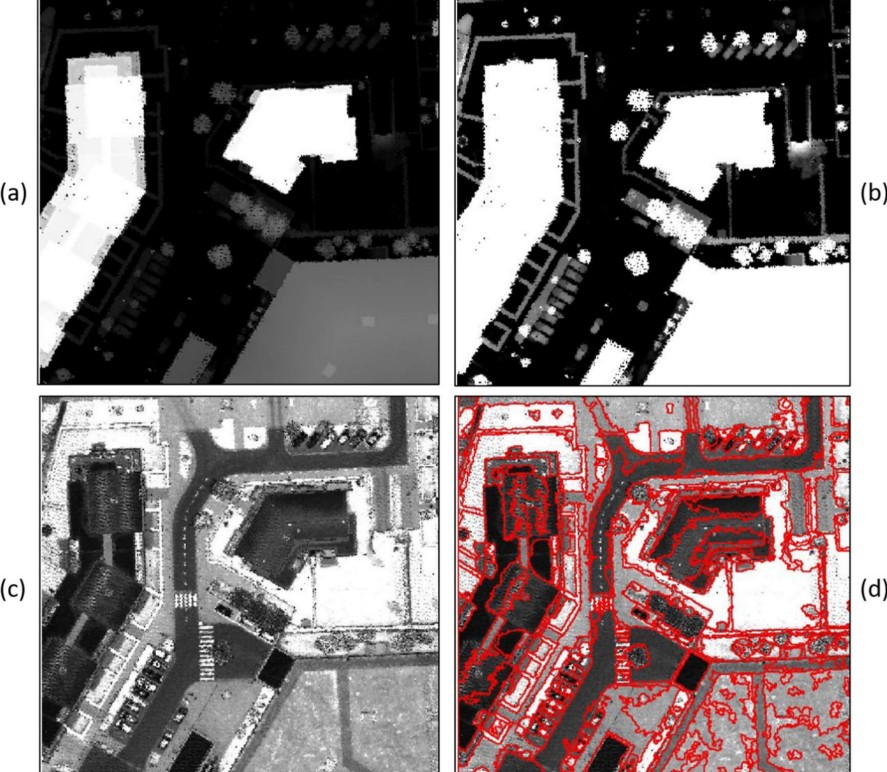

**Figure 4.** Example of input raster datasets used for image segmentation in this study and the corresponding output for one of the twenty validation blocks, with (**a**) nDSM1, (**b**) nDSM2, (**c**) intensity and (**d**) segmentation result depicted on intensity raster. Whereas nDSM1 shows the main height differences between buildings, trees and ground, nDSM2 highlights small height variations of low objects, e.g., individual hedges and cars.

Aside from the actual height and brightness of objects, the internal variation of both features within an object also show potential for classification purposes (e.g., height of a building is more homogeneous compared to height of a single tree's canopy, small height variations of natural grasslands compared to lawns). In image analysis techniques, these features are referred to as image texture. Here, we calculated four textural features (entropy, sum entropy, variance and sum variance) based on three different LiDAR features (nDSM1, nDSM2 and intensity).

## 2.4. Creation of Image Objects—Image Segmentation

Image segmentation, or the process of combining image pixels to create relatively homogeneous, non-overlapping image objects, lies at the foundation of object-based image analysis approaches. In order to enhance the detection of small objects, segmentation in this study was only based on LiDAR features (nDSM1, nDSM2 and intensity), i.e., the dataset featuring the highest spatial resolution. We adopted the segmentation workflow proposed by [55], which is based on the i.segment algorithm in GRASS GIS. The algorithm's parameters were set for each training and validation block separately using the Unsupervised Segmentation Parameter Optimization method [93]. An example of segmentation inputs and resulting output is provided in Figure 4.

## 2.5. Extraction of Training and Validation Object Features

For each image object, the mean and standard deviation of all spectral, structural and textural features were calculated and extracted using the i.segment.stats algorithm in GRASS GIS [55]. Additionally, geometrical features related to the object size and shape were also calculated, i.e., area, perimeter and compactness.

## 2.6. Identifying Most Suitable Image Features for Plant Type Classification through Random Forest Models

Random forest (RF) is a machine learning approach increasingly being used in remote sensing applications due to its relative high accuracy and computational efficiency compared to other frequently used machine learning approaches [94]. Since our goal was to compare the potential of different image datasets to label individual image objects, multiple RF models were trained, each based on a distinctive set of object features (Table A2). In addition, we also tested the benefits of a hierarchical classification approach, in which separate RF models were constructed and combined for subsequently differentiating vegetation from non-vegetation, woody from non-woody vegetation and more detailed urban green types. We split our training dataset (containing a total of 2543 objects) into training (70%) and testing (30%) objects according to a stratified random selection procedure (see also Table A1). Training was done using the default value of 500 trees, a random selection of ten values for the mtry hyperparameter and a 10 times repeated 5-fold cross validation approach (based on [55]). The best RF model was selected based on the total and class-based accuracies acquired for the independent test set. Variable importance of individual input features was assessed by means of the mean decrease in prediction error after permuting each predictor variable (default in R caret package).

## 2.7. Application of the Best Model, Post-classification Procedure and Accuracy Assessment

Being able to correctly classify homogeneous objects is a prerequisite to, but does not suffice for, the production of spatially continuous classification maps. The best performing RF model (cf. previous section) was therefore applied to the entire set of validation blocks to create spatially continuous classification maps. These results were critically evaluated on a visual basis and some re-classification rules were defined using eCognition software to correct for the most obvious errors (as was also done by e.g., [57]; see Table A4 for more details). Final classification accuracies were determined by calculating a confusion matrix and associated accuracy statistics (caret package, R software). Due to the unbalanced validation dataset used in this study, balanced accuracy (scaling between 0 and 1) was selected to describe model performance for individual classes.

## 2.8. Spatial Configuration of Trees and Shrubs

After obtaining a detailed plant type map, further distinction between detailed tree and shrub functional types based on spatial configuration (e.g., tree rows versus solitary trees) was accomplished through an additional rule-based classification procedure, which is described in detail in Table 2. In essence, individual trees and shrub objects were merged together, after which the resulting objects were classified based on their size, shape and distance to other trees/shrubs.

**Table 2.** Overview of rule-based classification procedure used to distinguish different functional types of trees and shrubs based on their spatial configuration. Procedure developed and applied in eCognition software.

| Distinction Between … | Classification Rules |
|---|---|
| Shrubs and hedges | If shrub AND Asymmetry ≥ 0.8 AND width ≤ 2.5 m —> hedge<br>If shrub AND compactness > 5 and width (main line) < 2 m —> hedge |
| Group of shrubs and scrub patch | If shrub AND width ≥ 15 m —> scrub patch<br>Else —> group of shrubs |
| Tree patch, tree row, solitary tree connected and solitary tree isolated | If tree AND asymmetry ≥ 0.8 AND width < 30 m —> tree row<br>If tree AND area < 15 m$^2$ —> solitary tree<br>If tree AND area < 150 m$^2$ AND asymmetry < 0.3 —> solitary tree<br>If solitary tree AND distance to other trees > 15 m —> solitary tree isolated<br>Else —> tree patch |
| Detection of wood verges | Merge all trees and shrub classes together<br>If combined object has asymmetry ≥ 0.8 AND area > 150 m$^2$ AND relative contribution of both tree and shrub < 0.7 —> wood verge |

## 3. Results

### 3.1. Potential of Remote Sensing Data for Differentiating Functional Urban Green Types

As stated in Section 2.2.4, multiple Random Forest models were constructed in order to assess the potential of different image datasets for distinguishing functional urban green types. When targeting basic urban green classes, high class-wise accuracies (>0.8) were attained, irrespective of the image datasets being used, except for the soil (0.50–0.72) and agriculture (0.59–0.76) classes (Table 3a). Most of the classes could be mapped with adequate accuracy using just LiDAR data. Agriculture, extensive green roofs, soil and water were better discriminated upon adding spectral information, with hyperspectral data contributing more (respective increase by 21%, 18%, 22% and 18%) compared to multispectral data (14%, 7%, 6% and 0%). The approach of creating multiple models in a hierarchical classification approach only slightly benefited the classification accuracy of basic vegetation types (average increase of 1%), but had a clear positive effect for the soil (6%) and water (19%) classes. When considering detailed urban green classes, the differences in performance between the different image datasets became more pronounced. Total accuracy increased by respectively 3% and 8% upon adding multispectral and hyperspectral data to LiDAR data, with maximum increases per class for the latter amounting to 34% (evergreen coniferous tree) and 38% (arable land) (Table 3b). Despite the availability of detailed hyperspectral and LiDAR data, several detailed urban green types remained hard to distinguish (evergreen coniferous and broadleaf shrub, flower bed and vegetable garden featured maximum accuracies below 0.80). Whereas the non-hierarchical approach mostly favored the most abundant land cover classes in our dataset (e.g., lawn), the hierarchical approach resulted in a considerable improvement for some of the more uncommon classes (e.g., flower bed by 21%; evergreen coniferous tree and arable land both by 7%).

**Table 3.** Overview of the best overall and class-wise (balanced) accuracies attained for the object-based classification of individual test objects using three different data sources (APEX = hyperspectral + LiDAR; WV2 = multispectral + LiDAR; LiDAR = LiDAR only) in a hierarchical (H) and non-hierarchical (NH) classification approach. Results are shown for the classification of (**a**) basic (aggregated) urban green classes and (**b**) most detailed classes. Best accuracies are indicated in bold. More information on the specific object features used in each model is included in the Appendix A (Table A3).

| (a) BASIC CLASSES | | H-APEX | H-WV2 | H-LiDAR | NH-APEX | NH-WV2 | NH-LiDAR |
|---|---|---|---|---|---|---|---|
| Overall Accuracy | | 0.88 | 0.86 | 0.85 | **0.89** | 0.88 | 0.85 |
| Class-wise Accuracies | | | | | | | |
| 10 | Tree | **0.99** | 0.98 | 0.98 | **0.99** | **0.99** | 0.98 |
| 20 | Shrub | 0.90 | 0.90 | 0.92 | **0.93** | **0.93** | **0.93** |
| 30 | Herbaceous | **0.80** | **0.80** | 0.77 | 0.78 | 0.76 | 0.76 |
| 34 | Lawn | 0.92 | 0.92 | 0.90 | **0.93** | **0.93** | 0.90 |
| 40 | Agriculture | **0.76** | 0.71 | 0.59 | 0.74 | 0.66 | 0.59 |
| 50 | Ext. green roof | **0.80** | 0.70 | 0.70 | **0.80** | 0.70 | 0.60 |
| 60 | Roof | **0.98** | **0.98** | 0.96 | **0.98** | **0.98** | 0.96 |
| 70 | Pavement | **0.96** | 0.95 | 0.92 | **0.96** | **0.96** | 0.92 |
| 80 | Soil | **0.72** | 0.53 | 0.50 | 0.58 | 0.53 | 0.50 |
| 90 | Water | **1.00** | 0.83 | 0.83 | 0.83 | 0.67 | 0.67 |
| 100 | Cars | **0.94** | **0.94** | **0.94** | 0.93 | **0.94** | **0.94** |
| (b) DETAILED CLASSES | | H-APEX | H-WV2 | H-LiDAR | NH-APEX | NH-WV2 | NH-LiDAR |
| Overall Accuracy | | **0.81** | 0.76 | 0.74 | **0.81** | 0.77 | 0.75 |
| Class-wise Accuracies | | | | | | | |
| 10 | Deciduous broadleaf tree | **0.97** | 0.95 | 0.96 | **0.97** | 0.96 | 0.96 |
| 11 | Evergreen coniferous tree | **0.87** | 0.61 | 0.55 | 0.81 | 0.61 | 0.55 |
| 20 | Deciduous broadleaf shrub | 0.81 | 0.81 | **0.84** | 0.82 | 0.83 | **0.84** |
| 21 | Evergreen coniferous shrub | **0.61** | 0.53 | 0.51 | 0.59 | 0.57 | 0.52 |
| 22 | Evergreen broadleaf shrub | 0.73 | 0.71 | 0.71 | **0.78** | 0.69 | 0.72 |
| 31 | Tall herb vegetation | **0.80** | 0.78 | 0.77 | 0.74 | **0.80** | **0.80** |
| 32 | Flower bed | **0.68** | 0.63 | 0.63 | 0.54 | 0.50 | 0.54 |
| 33 | Meadow and flower field | **0.81** | 0.77 | 0.74 | 0.78 | 0.74 | 0.77 |
| 34 | Lawn | 0.92 | 0.92 | 0.90 | **0.93** | **0.93** | 0.92 |
| 40 | Arable land | **0.94** | 0.75 | 0.56 | 0.87 | 0.69 | 0.56 |
| 41 | Vegetable garden | 0.65 | **0.69** | 0.61 | 0.62 | 0.65 | 0.57 |
| 50 | Ext. green roof | **0.80** | 0.70 | 0.70 | **0.80** | 0.70 | 0.70 |
| 60 | Roof | 0.98 | 0.98 | 0.96 | **0.99** | 0.98 | 0.97 |
| 70 | Pavement | **0.96** | 0.95 | 0.92 | **0.96** | **0.96** | 0.92 |
| 80 | Soil | **0.72** | 0.53 | 0.50 | 0.64 | 0.53 | 0.50 |
| 90 | Water | **1.00** | 0.83 | 0.83 | 0.83 | 0.67 | 0.67 |
| 100 | Cars | 0.94 | 0.94 | 0.94 | 0.95 | 0.95 | **0.96** |

Regarding the specific object features to be used as input to the model, we observed that the 30 MNF bands derived from APEX data consistently outperformed the use of spectral indices based on the same data, as well as the 218 original APEX bands, except for mutually distinguishing the non-vegetation classes (see Appendix A, Table A3). The addition of LiDAR features consistently increased the classification accuracy, most notably in case of the more detailed urban green classes. Including textural and geometrical features on the other hand only increased model performance in some instances and only to a very limited extent. Based on the top five ranking of feature importance within the best performing Random Forest models, the most valuable object features for classification included nDSM1, nDSM2, treeIndex, APEX MNF band 2, LiDAR intensity, APEX MNF band 6, texture of nDSM2, followed by more APEX MNF bands.

### 3.2. Producing a Functional Urban Green Map

Based on the outcomes presented in Table 3, the hierarchical model using hyperspectral and LiDAR data was applied to all validation blocks to generate spatially continuous classification maps, one for each block. Overall accuracy of these initial maps was good, i.e., 0.86 for basic vegetation classes (Table 4a) and 0.84 for detailed classes (Table 4b), but mainly driven by the high coverage of relatively easily distinguishable classes like buildings, pavement, deciduous broadleaf trees and lawn. Amongst the basic classes, lowest class-wise accuracies were found for shrub (0.55), herbaceous vegetation (0.48) and soil (0.55). Aside from high mutual confusion between these classes, pavement and lawn turned out to be major sources of classification error for all three classes. With regard to the more detailed vegetation classes (Table 4b), evergreen coniferous trees were frequently classified as deciduous broadleaf trees, whereas detailed shrub and herbaceous vegetation classes featured even lower class-wise accuracies below 0.5 (Table 3b). Main sources of confusion for shrubs included broadleaf deciduous trees and mutual confusion between the three shrub classes, while more than half of the pixels labeled as either tall herb or flower beds were wrongly classified as meadows.

**Table 4.** Overall and class-wise (balanced) classification accuracies achieved after applying the best performing Random Forest model (cf. Table 3) on the twenty validation blocks ("initial classification"), after discarding zones with uncertainty higher than 0.7 and after applying a rule-based post-classification correction algorithm (cf. Table A4). n denotes the number of image pixels available per class. Due to the absence of agricultural lands in our validation dataset, both arable land and vegetable gardens have been omitted here.

| (a) BASIC CLASSES | Initial Classification | | Retaining Only Class Probability > 0.7 | | Post-Classification Correction |
|---|---|---|---|---|---|
| **Overall Accuracy** | **0.86** | | 0.94 | | 0.87 |
| **Kappa** | **0.82** | | 0.92 | | 0.84 |
| **Per class** | **Acc** | **n (×10$^3$)** | **Acc** | **Reduction n** | **Acc** |
| Tree | 0.90 | 754.0 | 0.94 | 0.07 | 0.93 |
| Shrub | 0.55 | 136.5 | 0.70 | 0.33 | 0.57 |
| Herbaceous | 0.48 | 99.1 | 0.82 | 0.34 | 0.52 |
| Lawn | 0.78 | 460.6 | 0.87 | 0.19 | 0.85 |
| Ext. green roof | 0.75 | 28.5 | 0.97 | 0.06 | 0.75 |
| Roof | 0.95 | 391.8 | 0.98 | 0.11 | 0.94 |
| Pavement | 0.91 | 1168.2 | 0.94 | 0.20 | 0.86 |
| Soil | 0.55 | 63.9 | 0.75 | 0.46 | 0.49 |
| Water | 0.92 | 96.9 | 0.99 | 0.09 | 0.96 |
| **Total** | | **3199.5** | | **0.17** | |
| (b) DETAILED CLASSES | Initial Classification | | Retaining Only Class Probabilities > 0.7 | | Post-Classification Correction |
| **Overall accuracy** | 0.84 | | 0.94 | | 0.86 |
| **Kappa** | 0.79 | | 0.92 | | 0.81 |
| **Per class** | **Acc** | **n (×10$^3$)** | **Acc** | **Reduction n** | **Acc** |
| DBT | 0.89 | 735.7 | 0.94 | 0.14 | 0.93 |
| ECT | 0.51 | 18.2 | 0.94 | 0.43 | 0.50 |
| DBS | 0.30 | 63.5 | 0.71 | 0.68 | 0.31 |
| ECS | 0.72 | 8.1 | 0.90 | 0.75 | 0.76 |
| EBS | 0.41 | 64.9 | 0.89 | 0.72 | 0.45 |
| Tall herb | 0.31 | 20.6 | 0.60 | 0.52 | 0.30 |
| Flower bed | 0.27 | 29.8 | 0.73 | 0.53 | 0.32 |
| Meadow & flower field | 0.26 | 48.6 | 0.51 | 0.41 | 0.28 |
| Lawn | 0.78 | 460.6 | 0.87 | 0.21 | 0.85 |
| Ext. green roof | 0.75 | 28.5 | 0.97 | 0.06 | 0.75 |
| Roof | 0.95 | 391.8 | 0.98 | 0.12 | 0.94 |
| Pavement | 0.91 | 1168.2 | 0.94 | 0.20 | 0.86 |
| Soil | 0.55 | 63.9 | 0.75 | 0.47 | 0.49 |
| Water | 0.92 | 96.9 | 0.99 | 0.09 | 0.96 |
| **Total** | | **3199.5** | | **0.21** | |

The Random Forest model not only produces a final classification label per image object, but also provides an indication of uncertainty by means of estimated class membership probabilities. By applying a simple threshold of < 0.7 to these probabilities, we explicitly mapped the location of objects being classified with high uncertainty (Figure 5; threshold chosen based on visual inspection of results). Aside from the confusion between detailed vegetation classes mentioned earlier, high classification uncertainty was primarily found near object borders (e.g., building edges classified as trees), in transition zones between two land covers (e.g., narrow pavement next to lawns modelled as lawns) and in shadowed areas (e.g., shadowed pavement classified as water or vegetation). These zones made up 17% and 21% of the total area to be classified respectively for the basic and detailed classification (Table 4). Discarding these uncertain zones from the accuracy assessment indeed considerably boosted classification performance up to 0.94 overall accuracy for both basic and detailed classification. Still, class-wise accuracies for detailed herbaceous vegetation classes (tall herb, flower bed and meadow), deciduous broadleaf shrub and soil remained rather low (≤0.75), indicating severe confusion between these particular classes.

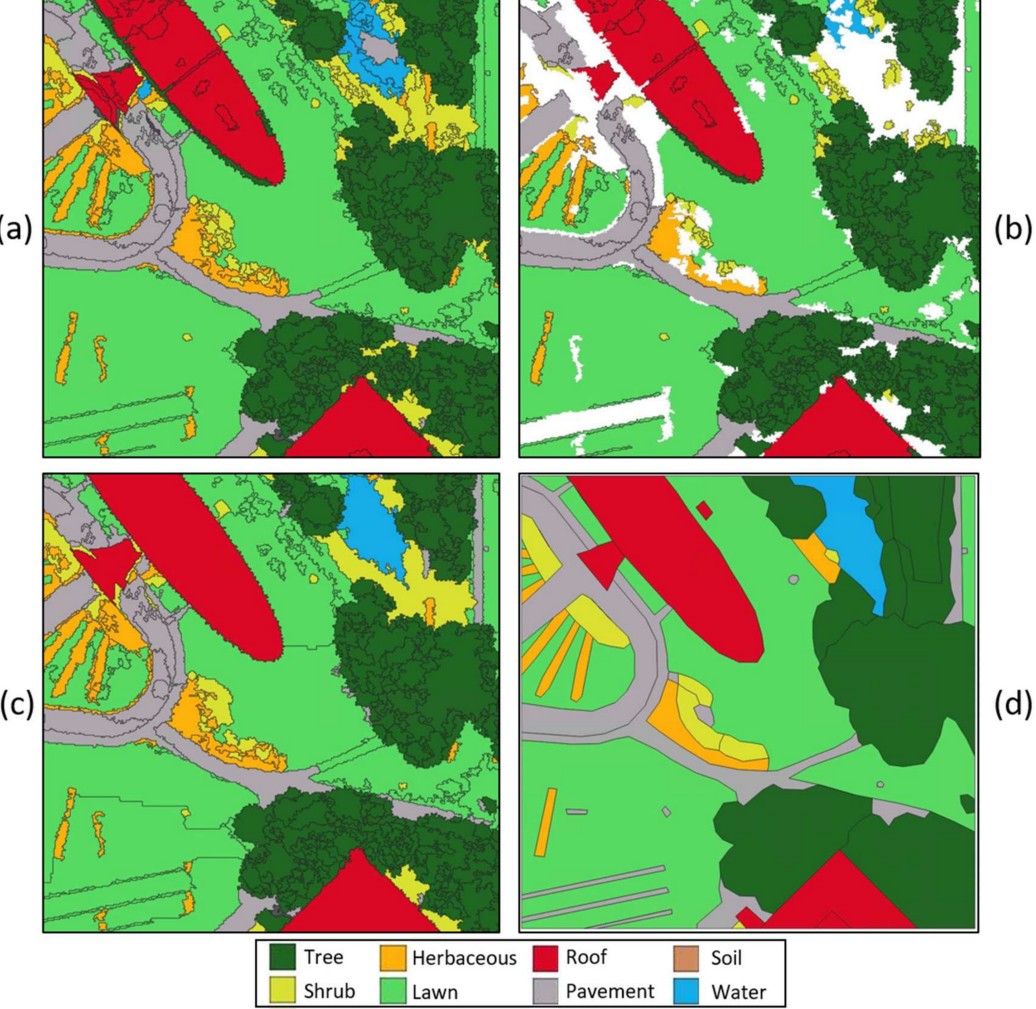

**Figure 5.** Example of classification results obtained for basic classes and for one out of twenty validation blocks (100 × 100 m), including (**a**) first classification result, (**b**) first classification result where areas featuring high classification uncertainty (class membership probability < 0.7) are masked out (white), (**c**) result after post-classification correction and (**d**) manually digitized reference data.

Instead of merely discarding these uncertain areas, we developed a rule-based post-classification procedure (Table A4), specifically aiming to reduce errors in zones affected by adjacency and/or

shadow effects. Due to its high spatial detail and being an active remote sensing technology, LiDAR data is inherently less prone to these disturbing effects and was hence mainly used during these post-classification corrections. A notable exception includes the water class, for which we used another water index, specifically designed to reduce the confusion with built-up surfaces in urban areas [95]. Although the net effect on overall classification accuracy was small, the proposed algorithm did increase the performance for all basic vegetation classes (mainly lawn) and water, reduced the accuracies for pavement and soil classes (Table 4a), but, more importantly, produced a classification map that visually made more sense (Figures 5 and 6). In particular, the detection of building edges was improved, thereby reducing confusion between roofs and trees, whereas pavement and soil were less frequently misclassified as low vegetation or water. Aside from reduced confusion between trees and shrubs, the post-classification procedure however did not enhance distinction between detailed urban green types (see Tables A5–A8 for a comparison between confusion matrices prior to and after post-classification correction).

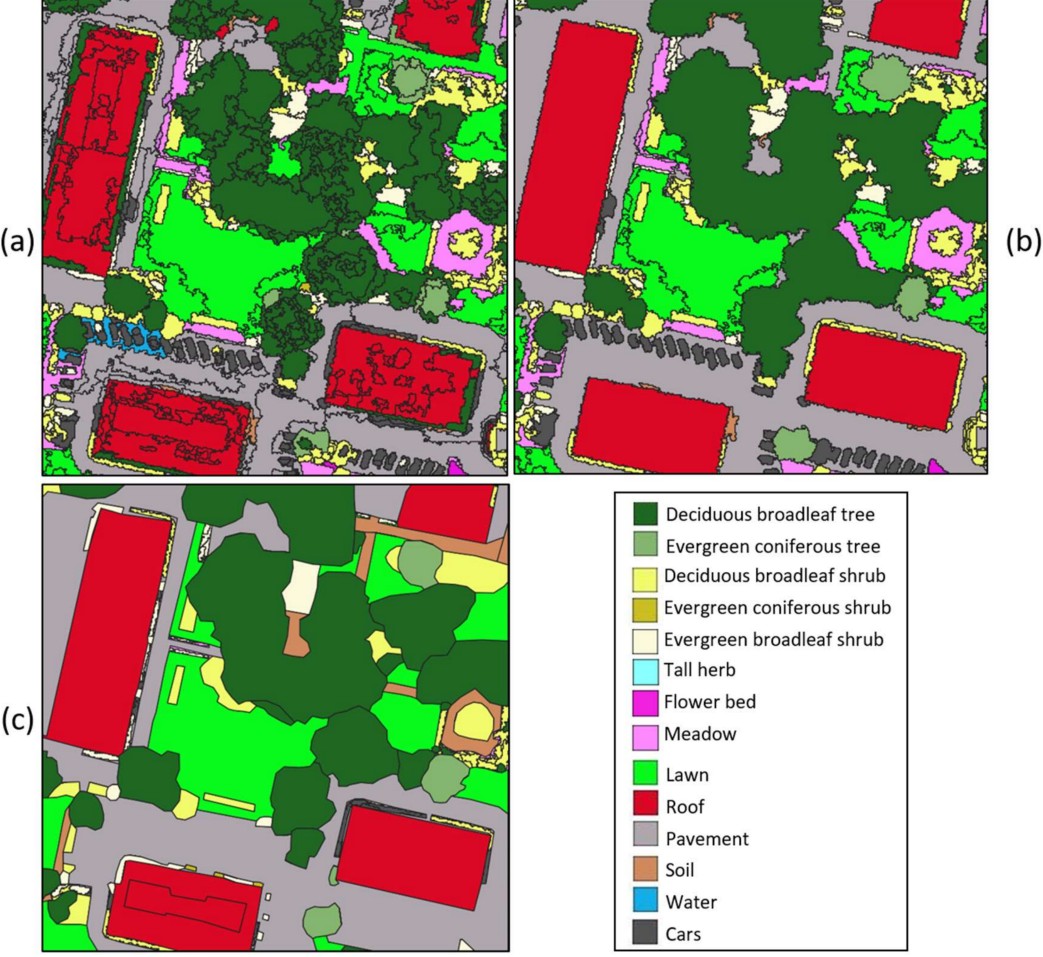

**Figure 6.** Example of detailed classification results obtained for one out of twenty validation blocks (100 × 100 m), including (**a**) initial classification result based on Random Forest model, (**b**) result after post-classification correction and (**c**) manually digitized reference data.

Finally, tree and shrubs were further classified based on their spatial configuration (cf. Table 2). As can be seen visually in Figure 7a, this simple procedure worked well for the distinction between narrow hedges and larger groups and patches of shrubs. Detection of tree rows on the other hand was not always successful, particularly in case the tree row directly interacted with a neighboring tree patch or when the crowns of individual trees within the row were not overlapping (Figure 7b,c).

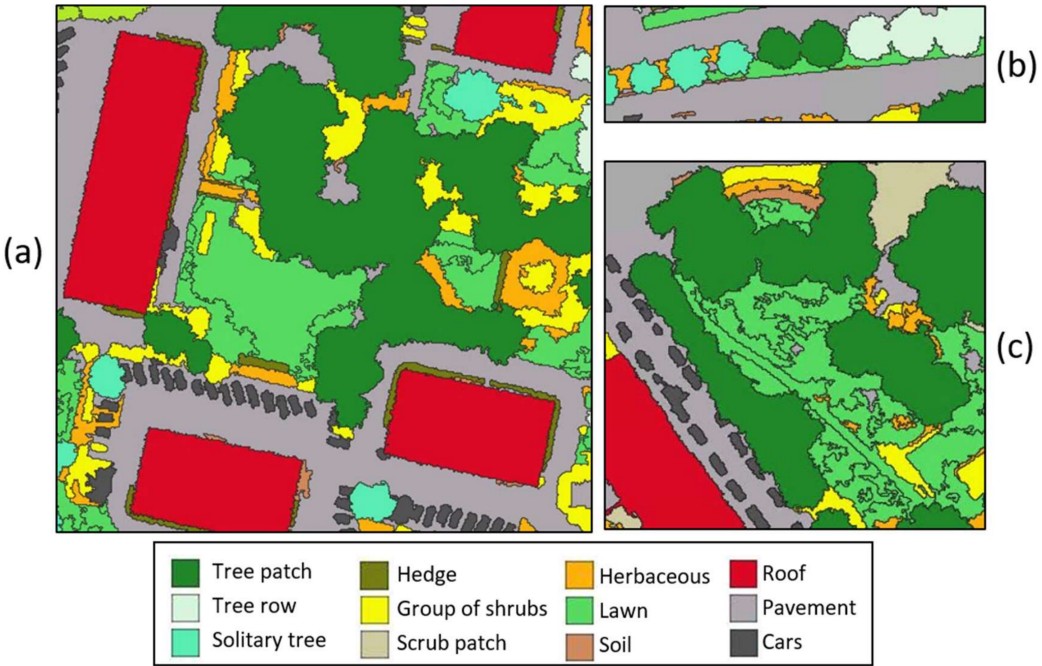

**Figure 7.** Detailed classification of trees and shrubs based on spatial configuration for (**a**) one entire validation block (cf. Figure 6), (**b**) one particular tree row, which was only partly labeled as tree row and (**c**) an area in which a tree row is in direct contact with a tree patch, causing the tree row not to be detected at all.

## 4. Discussion

### 4.1. Potential Applications of the Proposed Functional Urban Green Typology

Earlier research has already pointed to the need for uniform and spatially explicit datasets on urban green infrastructure within and across cities, in order to optimize the design and management of urban green spaces with regard to the provisioning of ecosystem services [96]. The functional urban green typology proposed here may act as a stepping stone towards accomplishing this goal. More specifically, the 23 functional urban green types can, in the first place, be used as a universal mapping framework to generate a detailed, spatially explicit view on urban ecosystem services through a value-transfer approach (cf. Section 1). Aside from a mapping methodology, which was the focus of the current paper, this approach requires a detailed ecosystem service scoring table indicating the relevance of the different urban green types for various ecosystem services. Whereas Table 1 already provides a qualitative starting point in this respect, more detailed quantitative ecosystem service scores would fully enable the use of the functional urban green typology in this sense. Some efforts have already been done to summarize the vast amount of scientific knowledge and empirical evidence on the link between urban green and ecosystem services. Derkzen, Van Teeffelen and Verburg [8] for instance published a list of six different ecosystem service indicator scores for seven, broad urban green types and used these to evaluate ecosystem services in Rotterdam (The Netherlands). Farrugia, Hudson and McCulloch [15] specifically focused on flood control and cooling and provided three related indicator values for 22 detailed urban green types. In 2015, the Flemish institute for technological research (VITO) published a report (in Dutch) on the valuation of ecosystem services in urban areas, including qualitative and quantitative ecosystem service scores covering eight ecosystem services and 43 urban green types based on an intensive literature review [97]. In turn, this report has been used as the basis for the Nature Value Explorer, an online tool allowing to calculate the implications of different (urban) planning scenarios on the provisioning of ecosystem services [98], and for the "Groentool", another

online tool designed for the city of Antwerp (Belgium) allowing to visualize the impact of different urban green scenarios on various ecosystem services [99].

Aside from a more holistic view on ecosystem services, the proposed typology may provide a solid framework to quantify particular urban ecosystem services in a detailed way using dedicated biophysical models, e.g., UrbClim [100] for urban heat fluxes and WetSpa [101] for urban water flows. Due to its intrinsic focus on ecosystem services, our functional typology provides more relevant classes compared to standard urban land cover products most commonly used as a basis for such models and most frequently generated by the urban remote sensing community, e.g., [36,37]. As a consequence, these models should be adapted to deal with the high thematic detail of the proposed typology. As different urban green characteristics might be relevant for different individual ecosystem services (e.g., leaf phenology for urban water and species information for ecological functions), the construction of a manageable typology that can directly be used to map each individual ecosystem service would be wildly impractical. Therefore, we would like to stress that the proposed urban green typology should be regarded as a flexible framework, which can be extended by additional information derived either from remote sensing (e.g., leaf phenology using multiple images acquired in different seasons [102]), additional spatial analysis (e.g., landscape connectivity [103]) or field inventories (e.g., detailed information on species or management practices) to meet the needs of the specific ecosystem service under consideration.

Although ecosystem services have been the main motivation behind our work and constitute the basis for the resulting functional green typology, this typology, together with the associated mapping workflow, could be adopted to serve many more applications. Indeed, such a spatially-explicit and detailed characterization of urban green represents essential information for urban green managers, allowing them to optimize their management activities across a city. Urban ecologists and environmentalists can use these detailed thematic maps to study interactions between the occurrence of certain urban green elements and the presence, abundance and reproduction potential of animal species, as well as several indicators for environmental quality (e.g., ambient temperature, air and soil pollution). This in turn will provide further insights into specific functions and services delivered by these urban green types. Finally, detailed maps on the composition of urban green within a city can provide valuable information to urban policymakers and planners on the current state, future priorities and desirable action points regarding urban green.

## 4.2. Mapping Functional Urban Green Types Using Remote Sensing Data

In the past, LiDAR data has been successfully applied to improve urban and/or vegetation classification performance by simply adding these data as a complementary data source to various approaches, by themselves mainly based on spectral information [40–42,51,52,54]. The results in this study however suggest that LiDAR data should take up a much more central role in detailed urban classification efforts. Not only does its high spatial resolution allows for detailed image segmentation (Figure 4), the various structural, spectral and textural features derived from LiDAR data were also found to be the most important classification features overall. This is in line with a study by Chen, Du, Wu, et al. [39], which concluded that height-related LiDAR features were more important compared to spectral features for urban land cover mapping. Whereas basic land cover classes could be readily differentiated using only LiDAR data, the added value of spectral data, and particularly of hyperspectral data, increased significantly when considering thematically more detailed urban (green) classes (Table 3). Conceptually, this can be explained by the higher degree of complementarity between the high spatial detail of LiDAR data on the one hand and the higher spectral information content in hyperspectral compared to multispectral data on the other hand, especially given the subtle spectral differences between different urban green types [27]. Likewise, the combination of hyperspectral data and LiDAR features was found to outperform combined multispectral and LiDAR data for detailed habitat mapping in Cumbria, UK [52]. New innovative ways are arising for combining hyperspectral

and LiDAR data in OBIA approaches (e.g., the concept of 3D hyperspectral point clouds [104]), opening up new and exciting possibilities for further research in this respect.

Both the adoption of a hierarchical classification approach and the application of dimensionality reduction techniques (in this case MNF) on the hyperspectral dataset improved classification accuracies for detailed urban green types (Table A3). These results agree with earlier findings regarding hierarchical classification of urban green [49] and relating to the added value of dimensionality reduction techniques for detailed vegetation classification [29,105]. Despite the use of spatially and spectrally detailed data sources and advanced analysis techniques (i.e., OBIA and Random Forest classification), uncertainties of detailed urban green types still remained high, particularly for shrub and herbaceous vegetation types (Table 3). Likewise, Mathieu, Aryal and Chong [21] reported only moderate accuracies of 63% up to 77% for detailed urban green mapping in the city of Dunedin, New Zealand, based on multispectral IKONOS imagery and OBIA techniques. Rather than merely using imagery acquired in summer (when the vegetation season is at its peak and all vegetation types appear green), as was done here, we highly suggest to further explore the potential of multi-temporal data for mapping these urban green types. As such, information regarding plant phenology can be integrated into the classification workflow, which is expected to benefit the distinction between evergreen and deciduous tree/shrub types and even individual species [30,106,107], between different herbaceous vegetation types [108] and between semi-natural versus agricultural land [109]. Yan, Zhou, Han, et al. [102] for instance found that phenology increased classification accuracies of broad urban green types by 10% to 13% when using an OBIA approach on Worldview-2 data.

Whereas classification performance on individual validation objects was not ideal but still acceptable (Table 3), accuracies considerably dropped when attempting to map spatially continuous areas (Table 4). Additional confusion was introduced particularly due to edge and adjacency effects, i.e. signal of one pixel affecting the signal of its neighboring pixels due to multiple scattering of light [110], and the high abundance of shadow (decreasing contrast, thereby making it harder to detect subtle spectral differences [111]), as could be derived from the spatial distribution of classification uncertainty (Figure 5b). Our rule-based post-classification procedure (Table A4) did resolve some visually evident misclassification errors (Figures 5 and 6), but did not lead to a significant improvement in overall accuracy (Table 4). One potential way to resolve this issue would be to collect additional training data, specifically targeting these edge and shadow regions. As the main goal of the current study was to assess the maximum potential of remote sensing data to differentiate various functional urban green types, we rather focused our efforts on collecting clear examples (i.e., pure and bright objects) of each functional type, which could explain the bad performance of our model in shadowed areas. These additional training data can then either be combined with all other training data in one and the same model, or can be used separately to train a specific model dedicated to classifying shadowed areas. Rather than merely labeling shadow as a separate class in land cover maps, as traditionally done by the urban remote sensing community [112,113], separately treating shadowed and non-shadowed areas in a hierarchical classification approach is becoming more and more common practice in order to reveal the true land cover composition of complex urban areas [42,47]. A second approach which could reduce the negative impacts of object edges and shadow is the concept of multi-scale or hierarchical segmentation, i.e., generating multiple, nested segmentation products for the same area using different scale parameters [114]. A careful selection of the most appropriate segmentation scale for each class of interest could lead to a more realistic representation of the complex urban landscape (e.g., selecting different scales for big buildings versus small hedges) and could effectively reduce the number of edge objects. Additionally, the use of features derived from multiple segmentation scales has been shown to significantly improve land cover classification performance over single segmentation approaches [115,116].

The remote sensing based mapping workflow presented here did certainly not cover all types or aspects of the proposed functional urban green typology (Table 1) in an equally detailed way. Particularly, the distinction between different tree and shrub functional types based on their spatial configuration

could be further improved using more in-depth contextual and spatial analysis (cf. Figure 7b,c), for instance based on specific metrics proposed by Wen, Huang, Liu, et al. [117] for semantic classification of urban trees (e.g., cohesion index, shape index, distance to road). Certain specific urban green types were not considered here due to their rarity in our study area and should be the focus of more, dedicated research (e.g., detection of water plants or intensive green roofs). Finally, due to its orientation, vertical green is not expected to be readily detectable using airborne remote sensing technology, stressing the need to look into complementary data sources, including Google Street View [118] or citizen science [119].

## 5. Conclusions

In this paper, we proposed a functional urban green typology and associated mapping workflow based on remote sensing data to facilitate the production of urban ecosystem service maps. The suggested typology, covering 23 functional types, may as such be used as a solid framework to produce a holistic view on urban ecosystem services through a simple value-transfer approach, but can also easily be extended using ancillary data for a more in-depth assessment of particular services. Our mapping workflow (comprised by a hierarchical, object-based random forest classification and subsequent rule-based post-classification correction) clearly demonstrated the potential, but also remaining limitations of remote sensing data for detailed urban green mapping. In general, airborne LiDAR data was found to be the most important data source for classification, but required complementary spectral data (preferably hyperspectral) when targeting urban green types at high thematic detail. The high spectral similarity between detailed urban green types and close interactions between different objects in the complex urban fabric (causing obscuring adjacency and shadow effects) were identified as the main sources of error, resulting in poor classification accuracies, especially for shrub and herbaceous vegetation classes (balanced accuracy <0.5). Nevertheless, we believe this work to provide a starting point for the further development of a functional urban green mapping workflow. In our opinion, the main focus for future research should be directed towards incorporating detailed information on phenology in the classification approach through the use of multi-temporal remote sensing data.

**Author Contributions:** Conceptualization, B.S., M.H. and J.D.; methodology, J.D. and B.S.; software, J.D.; validation, J.D. and B.S.; formal analysis, J.D.; investigation, J.D., B.S. and M.H.; resources, J.D.; data curation, J.D.; writing—original draft preparation, J.D.; writing—review and editing, B.S. and, M.H.; visualization, J.D.; supervision, B.S. and M.H.; project administration, B.S.; funding acquisition, B.S. and M.H. All authors have read and agreed to the published version of the manuscript.

**Funding:** The research presented in this paper is funded by the Belgian Science Policy Office in the framework of the STEREOIII program (UrbanEARS project (SR/00/307) and BelAir project (SR/01/354) for acquisition of hyperspectral data of Brussels).

**Acknowledgments:** The authors would like to express their gratitude to the Brussels Environmental Agency (BIM), and in particular to Mathias Engelbeen and Fabien Genard, for kindly providing Worldview-2 data and insights into the daily operation of urban green management in the city of Brussels. In addition, our thanks to Mike Alonzo for the interesting discussions on the general approach and specific methodology applied throughout this work. Finally, we would like to acknowledge Jingli Yan for his advice on object-based image processing and Joseph McFadden for our discussions regarding the functional urban green typology proposed in this study.

**Conflicts of Interest:** The authors declare no conflict of interest.

# Appendix A

**Table A1.** Sample size of training and testing datasets used for generating and testing Random Forest models in this study, compared to relative abundance of land cover classes in our twenty validation blocks.

| ID | Land Cover Class | Sample Sizes (Number of Objects) | | Relative Abundance Validation Blocks (%) |
|---|---|---|---|---|
| | | Training | Testing | |
| 10 | Deciduous broadleaf tree | 408 | 178 | 22.99 |
| 11 | Evergreen coniferous tree | 71 | 27 | 0.57 |
| 20 | Deciduous broadleaf shrub | 88 | 34 | 1.68 |
| 21 | Evergreen coniferous shrub | 18 | 12 | 0.04 |
| 22 | Evergreen broadleaf shrub | 63 | 34 | 1.31 |
| 31 | Tall herb vegetation | 38 | 18 | 0.14 |
| 32 | Flower bed | 28 | 11 | 0.93 |
| 33 | Meadow & flower field | 63 | 16 | 2.03 |
| 34 | Lawn | 142 | 59 | 14.41 |
| 40 | Arable land | 22 | 8 | 0.00 |
| 41 | Vegetable garden | 25 | 13 | 0.00 |
| 50 | Ext. green roof | 13 | 5 | 0.89 |
| 60 | Roof | 251 | 106 | 12.24 |
| 70 | Pavement | 240 | 105 | 36.51 |
| 80 | Soil | 25 | 18 | 2.00 |
| 90 | Water | 12 | 3 | 3.03 |
| 100 | Cars | 163 | 62 | 0.00 |

**Table A2.** Different combinations of object features used for training a random forest classification model. Each combination was tested in a hierarchical and non-hierarchical classification approach. For each feature (except for geometry features), both the object mean and standard deviation were included in the model.

| ID | Included Features | Number of Features |
|----|-------------------|--------------------|
| | **Hyperspectral** | |
| 1 | NDVI (apex) | 2 |
| 2 | APEX indices (NDVI, NDWI-G, NDWI-W, NDWI-M, GrassIdx, RedGreen ratio, BlueGreen ratio, brightness) | 16 |
| 3 | APEX indices + LiDAR features (nDSM1, nDSM2, intensity, treeIndex) | 24 |
| 4 | APEX indices + LiDAR features + texture features (texture of nDSM1, nDSM2, intensity) | 72 |
| 5 | APEX indices + LiDAR features + texture features + geometry (area, perimeter, compact_circle) | 75 |
| 6 | APEX bands (218 original spectral bands) | 416 |
| 7 | APEX bands + LiDAR features | 424 |
| 8 | APEX bands + LiDAR features + texture features | 472 |
| 9 | APEX bands + LiDAR features + texture features + geometry | 475 |
| 10 | APEX MNF (30 MNF transformed APEX bands) | 60 |
| 11 | APEX MNF + LiDAR features | 68 |
| 12 | APEX MNF + LiDAR features + texture features | 116 |
| 13 | APEX MNF + LiDAR features + texture features + geometry | 119 |
| | **Multispectral** | |
| 14 | NDVI (worldview-2) | 2 |
| 15 | NDVI + WV bands (all 8 original Worldview-2 bands) | 18 |
| 16 | NDVI + WV bands + LiDAR features | 26 |
| 17 | NDVI + WV bands + LiDAR features + texture features | 74 |
| 18 | NDVI + WV bands + LiDAR features + texture features + geometry | 77 |
| | **LiDAR only** | |
| 19 | LiDAR features | 8 |
| 20 | LiDAR features + texture features | 56 |
| 21 | LiDAR features + texture features + geometry | 59 |

**Table A3.** Total accuracies attained through object-based classification of individual test objects using different sets of object features as input to the classification algorithm. More information on the specific features used in each set is included in Table A2. Results are shown both for the hierarchical modelling approach and non-hierarchical model. In the former case, 7 individual models were created to distinguish (1) vegetation from non-vegetation, (2) woody (tree, shrub) from non-woody vegetation, (3) trees and shrubs, (4) detailed woody vegetation classes, (5) lawn, agriculture, extensive green roofs and other herbaceous vegetation, (6) detailed non-woody vegetation classes and (7) roof, pavement, soil and water. Two non-hierarchical models were produced, one for basic vegetation classes and one for most detailed vegetation classes. Highest accuracies are indicated in bold.

| Feature Set ID | Hierarchical Model | | | | | | | Non-hierarchical Model | |
|---|---|---|---|---|---|---|---|---|---|
| | **(1)** | **(2)** | **(3)** | **(4)** | **(5)** | **(6)** | **(7)** | **Basic** | **Detailed** |
| | **Hyperspectral + LiDAR** | | | | | | | | |
| 1 | 0.91 | 0.69 | 0.51 | 0.42 | 0.44 | 0.34 | 0.47 | 0.38 | 0.34 |
| 2 | 0.94 | 0.79 | 0.61 | 0.55 | 0.64 | 0.55 | 0.67 | 0.57 | 0.54 |
| 3 | 0.96 | 0.91 | 0.81 | 0.69 | 0.69 | 0.66 | 0.91 | 0.83 | 0.77 |
| 4 | 0.97 | **0.93** | 0.86 | 0.70 | 0.69 | 0.65 | 0.92 | 0.85 | 0.78 |
| 5 | 0.96 | 0.93 | 0.85 | 0.69 | 0.69 | 0.66 | 0.92 | 0.85 | 0.78 |
| 6 | 0.93 | 0.78 | 0.60 | 0.55 | 0.65 | 0.61 | 0.65 | 0.58 | 0.54 |
| 7 | 0.95 | 0.90 | 0.82 | 0.67 | 0.71 | 0.66 | 0.92 | 0.83 | 0.77 |
| 8 | 0.95 | 0.92 | 0.85 | 0.69 | 0.71 | 0.66 | **0.94** | 0.85 | 0.76 |
| 9 | 0.95 | 0.92 | 0.86 | 0.69 | 0.72 | 0.67 | 0.93 | 0.85 | 0.77 |
| 10 | 0.94 | 0.84 | 0.66 | 0.57 | 0.71 | 0.69 | 0.76 | 0.65 | 0.62 |
| 11 | 0.96 | 0.92 | 0.86 | **0.74** | 0.73 | 0.71 | 0.91 | 0.85 | **0.80** |
| 12 | 0.98 | 0.93 | 0.85 | 0.72 | **0.75** | **0.71** | 0.92 | **0.86** | 0.79 |
| 13 | **0.98** | 0.93 | **0.86** | 0.72 | 0.73 | 0.70 | 0.92 | 0.85 | 0.78 |
| | **Multispectral + LiDAR** | | | | | | | | |
| 14 | 0.90 | 0.65 | 0.54 | 0.44 | 0.47 | 0.38 | 0.39 | 0.34 | 0.30 |
| 15 | 0.92 | 0.75 | 0.67 | 0.52 | 0.57 | 0.51 | 0.54 | 0.53 | 0.50 |
| 16 | 0.95 | 0.90 | 0.83 | 0.63 | **0.72** | 0.64 | 0.88 | 0.82 | 0.73 |
| 17 | **0.96** | **0.92** | 0.85 | 0.67 | 0.69 | **0.65** | 0.89 | 0.83 | 0.76 |
| 18 | 0.96 | 0.92 | **0.86** | **0.69** | 0.70 | 0.63 | **0.90** | **0.83** | **0.76** |
| | **LiDAR Only** | | | | | | | | |
| 19 | 0.92 | 0.88 | 0.80 | 0.62 | 0.66 | **0.64** | 0.86 | 0.78 | 0.70 |
| 20 | 0.94 | **0.90** | 0.82 | 0.64 | 0.66 | 0.62 | **0.87** | **0.82** | 0.72 |
| 21 | **0.94** | 0.89 | **0.83** | **0.65** | **0.67** | 0.61 | 0.86 | 0.82 | **0.73** |

**Table A4.** Overview of rule-based post-classification procedure used to correct for visually obvious classification errors occurring after initial object-based classification of our twenty validation blocks. Procedure developed and applied in eCognition software.

| Source of Confusion / Error | Classification Rules |
|---|---|
| Shadowed areas wrongly classified as water | * the following is only applied to objects with class probability below 0.7 *<br>If NDWI_X > -0.3 —> water<br>Else if intensity > 0.3 AND NDVI > 0.6 —> herbaceous vegetation<br>Else —> pavement<br>If water AND area < 200 pixels AND NDVI > 0.6 —> lawn<br>If water AND area < 200 pixels AND NDVI ≤ 0.6 —> pavement |
| Water body wrongly classified as vegetation or pavement | * the following is only applied to objects with class probability below 0.7 *<br>If NDWI_X > -0.3 AND relative border to water > 0 —> water |
| Shaded or narrow pavement misclassified as vegetation | * the following is only applied to objects with class probability below 0.7 *<br>If herbaceous vegetation AND intensity < 0.4 AND NDVI < 0.85 —> pavement<br>If herbaceous vegetation AND NDVI < 0.2 —> pavement<br>If lawn AND NDVI < 0.2 —> pavement<br>If lawn AND intensity < 0.6 —> pavement<br>If cropland AND intensity < 0.23 —> pavement<br>If cropland AND intensity < 0.35 AND asymmetry > 0.84 —> pavement |
| Small patches classified as cropland | If cropland AND area < 600 pixels AND intensity ≥ 0.4 —> herbaceous vegetation<br>If cropland AND area < 600 pixels AND intensity < 0.4 —> soil |
| Cars classified as shrub | If shrub enclosed by car —> car<br>If shrub with relative border to car > 0.5 —> car<br>If shrub with relative border to car > 0.24 AND NDVI < 0.35 —> car<br>If shrub with relative border to car > 0.24 AND relative height difference < 0.13 —> car |
| Shrub classified as car | If car AND area < 52 pixels AND NDVI > 0.3 —> shrub<br>If car AND asymmetry > 0.8 AND NDVI > 0.2 —> shrub<br>If car AND compactness > 4 AND NDVI > 0.2 —> shrub |
| Roof edge misclassified as tree | If tree AND relative border to roof > 0.3 AND area < 500 pixels —> roof<br>If tree AND relative border to roof > 0.3 AND asymmetry > 0.95 —> roof<br>If tree fully enclosed by roof —> roof |
| Small portions of trees or shrubs classified as roof | If roof AND area < 200 pixels AND relative border to tree > 0.4 —> tree<br>If roof AND area < 200 pixels AND relative border to shrub > 0.4 —> shrub |
| Edges of trees misclassified as shrub (due to low height) | If shrub AND relative border to tree > 0.31 AND area < 300 pixels —> tree |
| Small parts of evergreen coniferous trees (ECT) misclassified as deciduous broadleaf trees (DBT) and other way around | If DBT AND relative border to ECT > 0.3 AND area < 400 pixels —> ECT<br>If ECT AND relative border to DBT > 0.3 AND area < 400 pixels —> DBT |

**Table A5.** Confusion matrix obtained for classifying all validation blocks according to the basic vegetation classes and using the best performing Random Forest model, i.e., a hierarchical model featuring hyperspectral and LiDAR data. Classification results are presented in the rows, reference classes in the columns. Red numbers indicate severe confusion (more than 5 % of the reference pixels of a certain class being classified as another class). Numbers marked in grey represent those confusions actively dealt with in the post-classification correction procedure.

| | Tree | Shrub | Herbaceous | Lawn | Crop-land | Ext. green roof | Roof | Pavement | Soil | Water | Total |
|---|---|---|---|---|---|---|---|---|---|---|---|
| Tree | 678,685 | 16,731 | 324 | 2620 | 0 | 408 | 26,503 | 7000 | 542 | 70 | 732,883 |
| Shrub | 26,777 | 90,576 | 8975 | 9852 | 0 | 0 | 2439 | 16,464 | 2179 | 2229 | 159,491 |
| Herbaceous | 8437 | 10,689 | 70,748 | 27,303 | 0 | 0 | 821 | 18,379 | 3743 | 1484 | 141,604 |
| Lawn | 15,599 | 6919 | 12,776 | 401,164 | 0 | 0 | 5506 | 38,844 | 14,082 | 421 | 495,311 |
| Cropland | 4066 | 298 | 585 | 1818 | 0 | 0 | 13 | 26,343 | 978 | 702 | 34,803 |
| Ext. green roof | 2 | 0 | 2 | 16 | 0 | 28,052 | 9117 | 80 | 0 | 0 | 37,269 |
| Roof | 991 | 1084 | 34 | 1453 | 0 | 28 | 342,389 | 11,587 | 1808 | 0 | 359,374 |
| Pavement | 17,311 | 9259 | 4878 | 12,491 | 0 | 0 | 4711 | 1,027,231 | 10,363 | 3677 | 1,089,921 |
| Soil | 1458 | 741 | 680 | 3065 | 0 | 0 | 243 | 17,114 | 30,026 | 0 | 53,327 |
| Water | 658 | 171 | 69 | 859 | 0 | 0 | 100 | 5130 | 191 | 88,357 | 95,535 |
| Total | 75,3984 | 136,468 | 99,071 | 460,641 | 0 | 28,488 | 391,842 | 1,168,172 | 63,912 | 96,940 | 3,199,518 |

**Table A6.** Confusion matrix obtained after applying a rule-based post-classification correction procedure on the results presented in Table A5. Red numbers indicate severe confusion (more than 5% of the reference pixels of a certain class being classified as another class). Numbers marked in grey represent those confusions actively dealt with in the post-classification correction procedure.

| | Tree | Shrub | Herbaceous | Lawn | Cropland | Ext. Green Roof | Roof | Pavement | Soil | Water | Total |
|---|---|---|---|---|---|---|---|---|---|---|---|
| **Tree** | 684,658 | 19,606 | 722 | 3763 | 0 | 408 | 5282 | 6521 | 755 | 210 | 721,925 |
| **Shrub** | 19,523 | 87,142 | 8569 | 8254 | 0 | 0 | 2597 | 16,968 | 2074 | 2089 | 147,216 |
| **Herbaceous** | 6811 | 9845 | 69,678 | 27,162 | 0 | 0 | 753 | 12,398 | 3333 | 847 | 130,827 |
| **Lawn** | 11,795 | 4880 | 9437 | 387,315 | 0 | 0 | 5270 | 15,810 | 8104 | 421 | 443,032 |
| **Cropland** | 510 | 46 | 27 | 142 | 0 | 0 | 13 | 6016 | 683 | 0 | 7437 |
| **Ext. green roof** | 2 | 0 | 2 | 16 | 0 | 28,052 | 9117 | 80 | 0 | 0 | 37,269 |
| **Roof** | 2288 | 1311 | 44 | 1879 | 0 | 28 | 363,529 | 13,986 | 1873 | 0 | 384,938 |
| **Pavement** | 25,533 | 12,839 | 9147 | 28,516 | 0 | 0 | 5025 | 1,071,473 | 16,783 | 3599 | 1,172,915 |
| **Soil** | 2702 | 782 | 934 | 3594 | 0 | 0 | 243 | 22,052 | 30,307 | 399 | 61,013 |
| **Water** | 162 | 17 | 511 | 0 | 0 | 0 | 13 | 2868 | 0 | 89,375 | 92,946 |
| **Total** | 753,984 | 136,468 | 99,071 | 460,641 | 0 | 28,488 | 391,842 | 1,168,172 | 63,912 | 96,940 | 3,199,518 |

**Table A7.** Confusion matrix obtained for classifying all validation blocks according to the most detailed vegetation classes and using the best performing Random Forest model, i.e., a hierarchical model featuring hyperspectral and LiDAR data. Red numbers indicate severe confusion (more than 5% of the reference pixels of a certain class being misclassified).

| | DBT | ECT | DBS | ECS | EBS | Tall herb | Flower bed | Meadow | Lawn | Arable land | Vegetable Garden | Ext. Green Roof | Roof | Pavement | Soil | Water | Total |
|---|---|---|---|---|---|---|---|---|---|---|---|---|---|---|---|---|---|
| **DBT** | 655,649 | 6106 | 10,641 | 109 | 5178 | 54 | 47 | 201 | 2518 | 0 | 0 | 408 | 26,193 | 6924 | 535 | 70 | 714,633 |
| **ECT** | 7562 | 9368 | 209 | 3 | 591 | 13 | 1 | 8 | 102 | 0 | 0 | 0 | 310 | 76 | 7 | 0 | 18,250 |
| **DBS** | 18,114 | 1078 | 29,566 | 1811 | 18,785 | 3058 | 624 | 2066 | 5387 | 0 | 0 | 0 | 1942 | 8917 | 1317 | 1117 | 93,782 |
| **ECS** | 280 | 0 | 159 | 2398 | 0 | 1 | 0 | 7 | 252 | 0 | 0 | 0 | 30 | 121 | 78 | 0 | 3326 |
| **EBS** | 7065 | 264 | 10,118 | 1626 | 26,114 | 1371 | 594 | 1254 | 4285 | 0 | 0 | 0 | 467 | 7500 | 784 | 1112 | 62,554 |
| **Tall herb** | 386 | 0 | 398 | 0 | 1014 | 2756 | 34 | 2859 | 351 | 0 | 0 | 0 | 62 | 266 | 119 | 643 | 8888 |
| **Flower bed** | 876 | 97 | 1673 | 215 | 916 | 552 | 5640 | 795 | 4462 | 0 | 0 | 0 | 290 | 4295 | 309 | 1 | 20,121 |
| **Meadow** | 6930 | 148 | 3467 | 458 | 2547 | 12,033 | 15,338 | 30,741 | 22,517 | 0 | 0 | 0 | 469 | 13,801 | 3315 | 840 | 112,604 |
| **Lawn** | 15,117 | 482 | 3316 | 366 | 3237 | 200 | 4319 | 8257 | 401,164 | 0 | 0 | 0 | 5506 | 38,844 | 14,082 | 421 | 495,311 |
| **Arable land** | 489 | 0 | 20 | 15 | 5 | 0 | 13 | 53 | 616 | 0 | 0 | 0 | 0 | 7414 | 108 | 47 | 8780 |
| **Vegetable garden** | 3530 | 23 | 29 | 48 | 181 | 14 | 258 | 247 | 1103 | 0 | 0 | 0 | 13 | 18,872 | 870 | 655 | 25,843 |
| **Ext. green roof** | 2 | 0 | 0 | 0 | 0 | 0 | 0 | 2 | 16 | 0 | 0 | 28,052 | 9117 | 80 | 0 | 0 | 37,269 |
| **Roof** | 845 | 146 | 511 | 64 | 509 | 0 | 12 | 22 | 1453 | 0 | 0 | 28 | 342,389 | 11,587 | 1808 | 0 | 359,374 |
| **Pavement** | 16,849 | 462 | 3169 | 794 | 5296 | 294 | 2584 | 2000 | 12,491 | 0 | 0 | 0 | 4711 | 1,027,231 | 10363 | 3677 | 1,089,921 |
| **Soil** | 1386 | 72 | 154 | 160 | 427 | 293 | 289 | 98 | 3065 | 0 | 0 | 0 | 243 | 17,114 | 30,026 | 0 | 53,327 |
| **Water** | 657 | 1 | 47 | 0 | 124 | 1 | 56 | 12 | 859 | 0 | 0 | 0 | 100 | 5130 | 191 | 88,357 | 95,535 |
| **Total** | 735,737 | 18,247 | 63,477 | 8067 | 64,924 | 20,640 | 29,809 | 48,622 | 460,641 | 0 | 0 | 28,488 | 391,842 | 1,168,172 | 63,912 | 96,940 | 3,199,518 |

DBT = Deciduous broadleaf tree; ECT = Evergreen coniferous tree; DBS = Deciduous broadleaf shrub; ECS = Evergreen coniferous shrub; EBS = Evergreen broadleaf shrub.

**Table A8.** Confusion matrix obtained after applying a rule-based post-classification correction procedure on the results presented in Table A7. Red numbers indicate severe confusion (more than 5% of the reference pixels of a certain class being classified as another class).

| | DBT | ECT | DBS | ECS | EBS | Tall herb | Flower bed | Meadow | Lawn | Arable land | Vegetable Garden | Ext. Green Roof | Roof | Pavement | Soil | Water | Total |
|---|---|---|---|---|---|---|---|---|---|---|---|---|---|---|---|---|---|
| DBT | 658,717 | 3622 | 11,471 | 300 | 6256 | 139 | 47 | 509 | 3605 | 0 | 0 | 408 | 5280 | 6291 | 706 | 206 | 697,557 |
| ECT | 9968 | 12,351 | 350 | 1 | 1228 | 0 | 11 | 16 | 158 | 0 | 0 | 0 | 2 | 230 | 49 | 4 | 24,368 |
| DBS | 14,371 | 717 | 29,121 | 1718 | 18,598 | 3007 | 614 | 1826 | 4329 | 0 | 0 | 0 | 2122 | 11,537 | 1436 | 1095 | 90,491 |
| ECS | 218 | 0 | 159 | 2187 | 0 | 1 | 0 | 7 | 198 | 0 | 0 | 0 | 18 | 24 | 72 | 0 | 2884 |
| EBS | 4009 | 208 | 9240 | 1499 | 24620 | 1350 | 594 | 1170 | 3727 | 0 | 0 | 0 | 457 | 5407 | 566 | 994 | 53,841 |
| Tall herb | 256 | 0 | 350 | 0 | 974 | 2331 | 34 | 2806 | 351 | 0 | 0 | 0 | 13 | 96 | 119 | 245 | 7575 |
| Flower bed | 580 | 96 | 1449 | 215 | 882 | 552 | 5636 | 589 | 3989 | 0 | 0 | 0 | 288 | 2962 | 199 | 1 | 17,438 |
| Meadow | 5732 | 147 | 3403 | 521 | 2051 | 12,013 | 15,245 | 30,472 | 22,822 | 0 | 0 | 0 | 452 | 9340 | 3015 | 601 | 105,814 |
| Lawn | 11,435 | 360 | 2357 | 288 | 2235 | 181 | 4137 | 5119 | 387,315 | 0 | 0 | 0 | 5270 | 15,810 | 8104 | 421 | 443,032 |
| Arable land | 95 | 0 | 13 | 0 | 5 | 0 | 5 | 0 | 25 | 0 | 0 | 0 | 0 | 1682 | 42 | 0 | 1867 |
| Vegetable garden | 415 | 0 | 3 | 0 | 25 | 0 | 0 | 22 | 117 | 0 | 0 | 0 | 13 | 4334 | 641 | 0 | 5570 |
| Ext. green roof | 2 | 0 | 0 | 0 | 0 | 0 | 0 | 2 | 16 | 0 | 0 | 28,052 | 9117 | 80 | 0 | 0 | 37,269 |
| Roof | 2224 | 64 | 433 | 72 | 806 | 0 | 12 | 32 | 1879 | 0 | 0 | 28 | 363,529 | 13,986 | 1873 | 0 | 384,938 |
| Pavement | 24,923 | 610 | 4974 | 1106 | 6759 | 738 | 2593 | 5816 | 28,516 | 0 | 0 | 0 | 5025 | 1,071,473 | 16,783 | 3599 | 1,172,915 |
| Soil | 2630 | 72 | 154 | 160 | 468 | 293 | 418 | 223 | 3594 | 0 | 0 | 0 | 243 | 22,052 | 30,307 | 399 | 61,013 |
| Water | 162 | 0 | 0 | 0 | 17 | 35 | 463 | 13 | 0 | 0 | 0 | 0 | 13 | 2868 | 0 | 89,375 | 92,946 |
| Total | 735,737 | 18,247 | 63,477 | 8067 | 64,924 | 20,640 | 29,809 | 48,622 | 460,641 | 0 | 0 | 28,488 | 391,842 | 1,168,172 | 63,912 | 96,940 | 3,199,518 |

DBT = Deciduous broadleaf tree; ECT = Evergreen coniferous tree; DBS = Deciduous broadleaf shrub; ECS = Evergreen coniferous shrub; EBS = Evergreen broadleaf shrub

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
