# Peer review of "Mapping Functional Urban Green Types Using High Resolution Remote Sensing Data"

_sustainability, doi:10.3390/su12052144_

Round 1

Reviewer 1 Report

I would like to the thank the author(s) for what was an informative read. My background is based in geosciences, and whilst slightly oblique to this field, I have enjoyed reviewing this manuscript. My qualifications for reviewing lie in my use of similar datasets (LiDAR and remote sensing) and methods (OBIA and machine learning) within geoscience and therefore I have focused my review on these areas. I hope you find these comments helpful and if they come across as impertinent, they are absolutely not meant to do so.

One overarching comment regarding most figures would be the accessibility of colour schemes. Whilst not a requirement of my review for all figures, I think any revisions based on this would be favourable to the reader and not be overly time-consuming for the author. Please find my specific comments below:

Line 72 - RPAS is perhaps a little unusual compared to the more common used term UAV (unmanned aerial vehicle). Ultimately, the author has defined the abbreviation but may wish to consider using UAV so this is concurrent with similar articles and may show up in associated searches.

Line 80 - the colours in the figure are difficult to distinguish, to increase accessibility consider using a 'colorbrewer' website to re-colour the linework

Line 89 - due to urban mapping of trees using OBIA and the random forest algorithm; Puissant et al. (2014) is worth acknowledging and incorporating where appropriate in other parts of the article.

Puissant, A., Rougier, S., & Stumpf, A. (2014). Object-oriented mapping of urban trees using Random Forest classifiers. International Journal of Applied Earth Observation and Geoinformation, 26, 235–245. https://doi.org/10.1016/j.jag.2013.07.002

Line 188 - figure 2, again consider the accessibility of the colours in this figure, e.g. green and yellow are difficult to distinguish

Line 217 - Section 2.2.3, the nomenclature here is muddled here. The author initially starts by describing validation blocks on line 218 but at the end of the paragraph these validation blocks are used for training (line 225). This is confusing and needs clarifying. From a technical point of view the author should include how many random samples were acquired, how many in each class (overall) and is this approximately proportional to what has been observed in the 'blocks' - assuming these are representative of the whole study area. Also, a figure illustrating the composition of these blocks would be a useful addition but perhaps better in the supplementary information.

Line 226 - I do not understand the term 'calibration blocks' in this context. Are they training blocks? I think the reader would benefit from a simplification of the wording here. From a technical point of view, the same should be applied as in the previous comment regarding number and proportion of samples etc especially as this was non-random and a brief comparison of the two sets is essential to determine any apparent bias imparted by the user - and compare the number/size of samples used (class balances etc).

Line 242 - the author uses a complex interaction of different softwares which makes reproducibility more difficult - would it be possible to indicate the use of different software in Figure 2?

Line 319 - can the author clarify if the reclassification is conducted on the 'best model' or on the preliminary map generated using the 'best model'? If the former, the confusion matrix and accuracies for the 'best model' prior to reclassification in post-processing must be reported. This can be reported in the Supplementary Information but obvious changes should be discussed in the manuscript.

In section 3 (results), the use of decimals and percentages are used interchangeably to describe accuracies, please standardise the usage here.

Line 342 the use of 'significant' here could imply a statistical significance that can be calculated - the author should either report the significance test used, or reword. 

Line 369-372 - what variable importance metric has been used here? There are several and worth clarifying as some are based on model accuracy and some are based on Gini (splitting). Also, reporting these numerically or, ideally, graphically would be advantageous - does one variable dominate over others?

Line 384 - The author uses kappa as a class-wise accuracy (Table 3b)? The reviewer was under the impression kappa is an overall statistic (which the author quotes as an overall statistic in Table 4). This in turn raises the question of what the author has used for describing class-wise accuracies which are generally referred to by two metrics, Producer Accuracy and User Accuracy or (precision and recall). Would the author be able to clarify/amend as required any confusion here. Also, reference Line 574.

Line 394-401 - the reviewer is particularly impressed with the thoughtfulness used by the author here but asks why a value of 0.7 was chosen to determine certainty over any other value (e.g. 0.8)? Is there a precedent to be referred to here?

Line 473 - perhaps this is an editorial comment but the use of 'obviously' is generally not advised in a discussion as it may impart

Line 575 - the reviewer would disagree, your study is better than 'decent' and you should not undermine your work through modesty

Line 576 - As a suggestion, "In our opinion, the main focus for future research...", may read better

Reviewer 2 Report

This is a very well-written paper, and it makes an important contribution to the literature on monitoring urban ecosystem services with remote sensing. I have only a few minor suggestions, outlined below:

Abstract

Page 1, Line 13: "Current mapping approaches however largely focus..." should be changed to "Current mapping approaches, however, largely focus..."

Page 1, Line 20: What does "class-wise accuracies < 0.5" mean? 50 percent? Please consider revising this sentence to more clearly convey the accuracy.

Introduction

I like the first two paragraphs of the introduction a lot. They clearly and compellingly introduce the importance of urban green space, and the mechanics behind using remote sensing to document urban green space.

Materials and Methods

Page 8, Line 225-226: How many objects were randomly sampled from each block to serve as the training data?

Reviewer 3 Report

This article is clear and well written, I like how it is organized, it is well introduced, with a complete review of references. Explanations, methodology, dataset, experiments, techniques, and results are convincing and interesting. It is remarkable how this research has been carefully done. My main concern is that this work comes directly from a copy&paste made from a PhD dissertation that it is not even referenced on this article when figures, tables, most text and results are exactly the same as the ones used in the PhD dissertation. I am not personally against publishing your PhD dissertation in a journal, but I wonder if the statement found on the PhD publication “All rights reserved. No part of the publication may be reproduced in any form by print, photoprint, microfilm, electronic or any other means without written permission from the publisher.” Should be addressed before publishing this paper. Apart from this fact, the paper is very good and congratulations for the PhD research.

Minor comments:

It is not required at all, but I suggest changing 2 m and 0.5 m in the abstract with 2 meters and 0.5 meter (just for trying to avoid acronyms in the abstract).

Reviewer 4 Report

The article focuses on mapping functional urban vegetation types, using various remote sensing data, from multi-spectral to hyper-spectral and LiDAR. It presents in detail a mapping method using the most up-to-date data sources and image processing techniques. I recommend the article to be accepted for publication after minor revision, respecting the following suggestions.

The proposed mapping approach can be used with various purposes, among which, the authors highlight “detailed assessment of urban ecosystem services” (line 105). Nonetheless, the authors do not actually assess the value of the services, explaining why they preferred not to do so (lines 474-482). Accepting the authors’ arguments, I suggest removing “ecosystem services” from the article objectives, and extending the Discussion section, by discussing possible applications of the proposed mapping approach adding other possible uses, in addition to ecosystem services.

Table 1: Although, the authors discuss relatively well the proposed functional urban green typology, they should provide more details on how they differentiated between the contributions: important, low and no contribution. In addition, it would be desirable if they justify the thresholds used: 3-15 m for trees planted at regular intervals, a group of shrubs of less than 15 m wide etc. Why these numbers we used? What do the star signs and superscript numbers in Table 1 mean? There should be a note attached to the table with explanations.

Overall, the method is presented very good. There are 13 pages out of 20 (without Appendix) representing step-by-step description of the image processing workflow. I particularly like Table 3, which quantifies accuracies depending on the data used.

Some technical observations:

There are errors in page numbering, which start over after each section break.

Sources of figures and tables should be explicitly shown. For example, it is not clear Figure 1 is the own elaboration of the authors, or they took it from somewhere.

The article focuses on mapping functional urban vegetation types, using various remote sensing data, from multi-spectral to hyper-spectral and LiDAR. It presents in detail a mapping method using the most up-to-date data sources and image processing techniques. I recommend the article to be accepted for publication after minor revision, respecting the following suggestions.

The proposed mapping approach can be used with various purposes, among which, the authors highlight “detailed assessment of urban ecosystem services” (line 105). Nonetheless, the authors do not actually assess the value of the services, explaining why they preferred not to do so (lines 474-482). Accepting the authors’ arguments, I suggest removing “ecosystem services” from the article objectives, and extending the Discussion section, by discussing possible applications of the proposed mapping approach adding other possible uses, in addition to ecosystem services.

Table 1: Although, the authors discuss relatively well the proposed functional urban green typology, they should provide more details on how they differentiated between the contributions: important, low and no contribution. In addition, it would be desirable if they justify the thresholds used: 3-15 m for trees planted at regular intervals, a group of shrubs of less than 15 m wide etc. Why these numbers we used? What do the star signs and superscript numbers in Table 1 mean? There should be a note attached to the table with explanations.

Overall, the method is presented very good. There are 13 pages out of 20 (without Appendix) representing step-by-step description of the image processing workflow. I particularly like Table 3, which quantifies accuracies depending on the data used.

Some technical observations:

There are errors in page numbering, which start over after each section break.

Sources of figures and tables should be explicitly shown. For example, it is not clear Figure 1 is the own elaboration of the authors, or they took it from somewhere.

Round 2

Reviewer 3 Report

I have read your reply and:

(i) it is normal procedure for PhD research to be published in a peer-reviewed journal, as well as in a PhD manuscript, without this being mentioned in the journal article, => well, I would not say it is normal procedure, in my opinion is all the way around, and even more when there is so much overlap. But the right question is: if this article is fully based on something published before, with same paragraphs, tables and figures...it is right not mention and reference the research source? I have a clear answer, what is yours?

(ii) the PhD manuscript specifically contains a statement that the work is being prepared for submission to a peer-reviewed journal: => that is fine, but again, no reference at all

(iii) PhD manuscripts are seldomly being distributed or read by other experts in the field, and we felt it would be a missed opportunity in case this research would not be published in a more accessible format (based on the review reports, the four reviewers tend to agree with this statement): => I am completely fine with publishing papers from the PhD dissertation, but against not including the reference to the original work.

(iv) the presented manuscript is not a direct copy & paste from the PhD manuscript, but was carefully restructured to better fit the format of a journal article. => I can see that work, but several paragraphs, tables and figures are exactly the same without any change, that is a fact.